# Co-administration of AYUSH 64 as an adjunct to standard of care in mild and moderate COVID-19: A randomized, controlled, multicentric clinical trial

**Arvind Chopra** [1] *, **Girish Tillu**[2], **Kuldeep Chuadhary** [3], **Govind Reddy**[4], **Alok Srivastava**[5], **Muffazal Lakdawala**[6], **Dilip Gode**[7], **Himanshu Reddy**[8], **Sanjay Tamboli**[9], **Manjit Saluja**[1], **Sanjeev Sarmukaddam** [1], **Manohar Gundeti**[3], **Ashwini Kumar Raut**[10], **B. C. S. Rao**[11], **Babita Yadav**[11], **Narayanam Srikanth**[11], **Bhushan Patwardhan**[2]

**1** Centre for Rheumatic Diseases, Pune, India, **2** Interdisciplinary School of Health Sciences, Savitribai Phule Pune University, Pune, India, **3** Central Ayurveda Research Institute, Mumbai, India, **4** Regional Ayurveda Research Institute, Nagpur, India, **5** Regional Ayurveda Research Institute, Lucknow, India, **6** H.N. Reliance Foundation Hospital and Research Centre, Mumbai, India, **7** Datta Meghe Institute of Medical Sciences, Nagpur, India, **8** King George's Medical University, Lucknow, India, **9** Target Institute of Medical Education & Research, Mumbai, India, **10** Medical Research Centre, Kasturba Health Society, Mumbai, India, **11** Central Council for Research in Ayurvedic Sciences, New Delhi, India

* arvindchopra60@hotmail.com

## Abstract

### Objective

Evaluate the efficacy of AYUSH 64, a standard polyherbal Ayurvedic drug in COVID-19.

### Methods

During the first pandemic wave, 140 consenting and eligible hospitalized adult participants with mild-moderate symptomatic disease (specific standard RT-PCR assay positive) were selected as per a convenience sample, and randomized (1:1 ratio) to an open-label (assessor blind) two-arm multicentric drug trial; standard of care (SOC as per Indian guidelines) versus AYUSH 64 combined with SOC (AYUSH plus). Participants were assessed daily and discharged once clinical recovery (CR, primary efficacy) was achieved which was based on a predetermined set of criteria (resolution of symptoms, normal peripheral oximetry, and negative specific RT-PCR assay). Each participant was followed using an indigenous software program(mobile phone) and completed a 12-week study period. The dose of AYUSH 64 was 2 tablets oral, 500 mg each, bid for 12 weeks (AYUSH plus only). Significant P was <0.05 (two-sided). On randomization, the groups were found well matched.

### Results

The mean interval time from randomization to CR was significantly superior in the AYUSH plus group [mean 6.45 days versus 8.26 days, 95% Confidence Interval of the difference -3.02 to -0.59 (P = 0.003, Student's 't test] as per-protocol analysis (134 participants); significant (P = 0.002) on an intention to treat analysis. 70% of the participants in AYUSH plus

**Data Availability Statement:** Several relevant data are within the manuscript and its Supporting Information files (S8F).All relevant data on primary

efficacy, adverse events and withdrawals is available enclosed with the manuscript as a Supporting Information file (S8 File). The full data set is archived by the CCRAS, Government of India (http://www.ccras.nic.in), as an electronic data base and access will be provided after approval by Director General, CCRAS as explained below. Access to full data set will be available for any kind of non-commercial purpose and without any other restriction after approval by the sponsor (Director General, Central Council for Research in Ayurvedic Sciences/ CCRAS, Government of India) six months after publication. The latter will require a formal request from the applicant stating the reason for access and accompanied with a CV (applicant) and may be sent Email (crdp5624@gmail.com) to Arvind Chopra (first author) for further processing by CCRAS.

**Funding:** The current study was sponsored by Central Council of Research in Ayurvedic Sciences (CCRAS), Ministry of AYUSH, Government of India. CCRAS authorized the study grant vide their Order Reference F.No.3-61/2020-CCRAS/Admn/IMR/458 dated 02 June 2020 (CCRAS website: http://www. ccras.nic.in). The grant was distributed and supervised by the authorized CCRAS officer to the 3 CCRAS run study sites. No individual was paid any part of the research grant. CCRAS selected the principal investigators from the study sites. The PIs selected the study site staff who were paid salary/ compensation from the site grant as a-priori approved by the CCRAS. CCRAS also appointed a 'Contract research organization (CRO)' on contract payment to supervise and co-ordinate the trial as per the Government policy. CCRAS did not play any role in the study design, data collection and analysis, decision to publish, or preparation of the manuscript. AYUSH 64, the investigational product in this study, was a proprietary formulation of CCRAS and directly (central procurement) supplied to the study sites. None of the authors received any funding for participation in the current study project. Amongst the authors, KC, GR, AS, ML,DG, HR, MG, BCSR, BY, NS were Ayurvedic physician investigators and salaried employees of CCRAS run Government medical institutions. ST was paid by the CRO. AR and GT were Ayurvedic physician consultants and AR was paid a honorarium by the CCRAS. AC was a rheumatologist in practise and appointed as the Chief Clinical Coordinator of AYUSH CSIR Project (research drug trials in COVID-19). BP was the Chairman, Interdisciplinary AYUSH R & D Task Force on COVID-19 set up Ministry of AYUSH, Government of India. MS was a research coordinator and assisted AC. AC, MS and BP worked in a voluntary capacity and did not receive any remuneration from CCRAS.

recovered during the first week (P = 0.046, Chi-square) and showed a significantly better change in physical health, fatigue, and quality of life measures. 48 adverse events, mostly mild and gut related, were reported by each group. There were 20 patient withdrawals (8 in AYUSH plus) but none due to an AE. There were no deaths. Daily assessment (hospitalization) and supervised drug intake ensured robust efficacy data. The open-label design was a concern (study outcome).

## Conclusions

AYUSH 64 in combination with SOC hastened recovery, reduced hospitalization, and improved health in COVID-19. It was considered safe and well-tolerated. Further clinical validation (Phase III) is required.

## Trial registration

CTRI/2020/06/025557.

## Introduction

The world continues to reel under the tragic burden of the COVID-19 pandemic. The medical system was precariously overstretched and scarred. Several drug trials were completed and many more are underway to unravel evidence-based medicine (EBM) for more effective and safe management [1]. However, despite several advances, the treatment of mild and moderate COVID-19 predominantly remains symptomatic and empirical, and data from drug trials is sparse [1–3].

It is prudent to state upfront that most of the patients of COVID-19 suffered from asymptomatic or mild and moderate disease and recovered without any complication or sequel [4, 5]. Sometimes the disease was rapidly progressive, and less than 10% of subjects reported severe disease [4–6]. An exuberant and dysregulated immune response was central to the progression and severity, life-threatening complications, and fatality [6, 7]. Several drugs were repurposed and extensively used for the chemoprophylaxis and treatment of COVID-19 [8]. The widespread use of hydroxychloroquine (HCQS) during the early pandemic in India was grossly restricted when drug trials failed to show unequivocal efficacy [9]. HCQS is no longer recommended [2, 3]. Despite limited evidence but based on good clinical experience, some drugs such as tocilizumab and remdesivir are still being used [2, 10, 11]. The use of steroids in severe disease became pivotal following the result of a single large, controlled drug trial [12].

The search to repurpose drugs (COVID-19) also rekindled vigorous research in the traditional, complementary, and alternative systems of medicine (CAM) [13, 14]. The potential for prophylaxis and treatment of COVID-19 in the Ayurveda medicinal system was encouraged by the popular use of several standard herbal drugs to treat febrile respiratory disorders, improve health and immunity since ancient times and the growing modern experimental evidence of their potent anti-inflammatory and immune modulation effects [15, 16]. Several medicinal plants were considered as potential therapeutic candidates [14, 17, 18]. Despite the limited scientific evidence, a large number of Indian population used Ayurvedic and other CAM drugs to prevent and treat COVID-19 from beginning of the pandemic [19, 20]. The deep-rooted belief in the safety and tolerability of Ayurvedic drugs was certainly an advantage [21].

**Competing interests:** The authors have declared that no competing interests exist. The authors declare their relationship related to study as per the International Committee of Medical Journal Editors is described in the section on 'Financial Disclosure' (see above). "This does not alter our adherence to PLOS ONE policies on sharing data and materials."

India has a legal system to regulate and promote plural systems of medicine including Ayurveda, Yoga, Naturopathy, Unani, Siddha, Sowa Rigpa, and Homoeopathy, which together are known as AYUSH systems. The Ministry of AYUSH established an Interdisciplinary AYUSH Research and Development Task Force on COVID-19 to promote scientific research and worked closely with the Ministry of Health and Family Welfare to manage and curb the pandemic [22, 23]. The Ministry of AYUSH and its research wing namely CCRAS (Central Council for Research in Ayurvedic Sciences) in collaboration with the Council of Scientific and Industrial Research (CSIR) also sponsored controlled drug trial studies in April 2020 to individually evaluate the therapeutic efficacy of 3 shortlisted Ayurvedic drugs as an adjunct to the standard of care (SOC) in the treatment of mild and moderate symptomatic COVID-19. Based on a common protocol, three drug trials were carried out at different sites with different investigators [24]. A controlled drug trial of AYUSH 64, a standard proprietary poly-herbal formulation of CCRAS, was amongst the latter studies.

The selection of AYUSH-64 was based on Ayurvedic logic and clinical experience. It was initially developed to treat malaria [25]. Later on, it was also found useful to treat cough and other mild respiratory tract infections and other disorders [18, 26]. Though readily available in AYUSH medical centers, its overall use remained limited. A comprehensive description of AYUSH 64 and other Indian medicinal plants with a therapeutic potential in COVID-19 was recently published [18].

The current report presents the results of the AYUSH-64 drug trial.

## Methods

The protocol was approved by the following Institutional Ethics Committee at each study site:-

1. Institutional Ethics Committee, Datta Meghe Institute of Medical Sciences, Nagpur (No. DMIMS(DU)/IEC/2020/8785)

2. Institutional Ethics Committee, Central Ayurveda Research Institute, Mumbai (No. 01/20-21)

3. Institutional Ethics Committee, King George's Medical University, Lucknow (No. 469/Ethics/2020).

The study was a prospective, randomized, open label (assessor blind), parallel efficacy, two arm multicentric drug trial. The protocol was registered with the Clinical Trials Registry of India (CTRI) (registration number CTRI/2020/06/025557) [24]. The protocol is enclosed as a S2 File. The study duration for each participant was 12 weeks. The study was carried out in the Government approved facilities for COVID- 19 in the medical and teaching hospitals at King George Medical University, Lucknow, Central Ayurveda Research Institute for Cancer, Mumbai, and Datta Meghe Institute of Medical Sciences, Nagpur. The study was carried out in accordance with the principles of Good Clinical Practice (GCP), Declaration of Helsinki (Brazil update 2013), ICMR (Indian Council of Medical Research), and CCRAS Guidelines (2018) [27, 28] The protocol and the study report also complied with CONSORT guidelines (a checklist is enclosed as S1 File) [29]. An independent data safety management board (DSMB) and a monitoring committee were appointed by the sponsor. An independent accredited CRO (Clinical Research Organization) was engaged by the sponsor for study oversight and regular monitoring checks, on-site training of personnel, implementation of study protocol, creation of a central study database and preparation of a study report.

The overall scheme of the study, study procedures, and predetermined time points of evaluation are shown in Fig 1.

INFORMED CONSENT & SCREENING OF VOLUNTARY IN-PATIENTS OF MILD AND MODERATE COVID-19 AS PER PROTOCOL; ELIGIBILITY AS PER INCLUSION-EXCLUSION CRITERIA

RANDOMIZE ELIGIBLE PATIENTS

STANDARD OF CARE (SOC) OR AYUSH PLUS (AYUSH 64 along with SOC)

| TIME POINTS | ASSESSMENT ‡ | INVESTIGATIONS |
|---|---|---|
| RANDOMIIZED BASELINE | HISTORY & CLINICAL EXAMINATION, AYURVEDA ASSESSMENT, ASSESS PHYSICAL & MENTAL HEALTH & QOL | SPECIFIC SARS-CoV-2 RT-PCR ASSAY, ROUTINE HEMATOLOGY, SPECIAL COVID-19 ASSAYS, X-RAY CHEST, EKG |
| RANDOMIZED TREATMENT PHASE (IN-PATIENT) | ASSESS CLINICAL STATUS & PULSE OXIMETRY DAILY TILL CLINICAL RECOVERY AND HOSPITAL DISCHARGE | REPEAT AS PER CLINICAL JUDGEMENT; REPEAT SPECIFIC RT-PCR ASSAY, X-RAY CHEST & EKG ON RECOVERY |
| WEEK 4 | CLINICAL EXAMINATION (INCUDING AYURVEDA) FOR AE & COVID-19 COMPLICATIONS, ASSESS PHYSICAL & MENTAL HEALTH & QOL | ROUTINE HEMATOLOGY, SPECIAL COVID-19 ASSAY; X-RAY CHEST (need basis) |
| WEEK 8 | REPEAT THE WEEK 4 ASSESSMENT | REPEAT WEEK 4 INVESTIGATIONS |
| WEEK 12 | REPEAT WEEK 4 ASSESSMENT & PERFORM COMPLETION ASSESSMENT | REPEAT WEEK 4 INVESTIGATIONS |

NOTE: (1) *SPECIAL MOBILE APP CALLED COVID KAVACH USED DAILY TO MONITOR SYMPTOMS AND AE FOLLOWING HOSPITAL DISCHARGE (2)ABBREVIATIONS:RT-PCR: REVERSE TRANSCRIPTASE POLYMERASE CHAIN REACTION ASSAY; AE: ADVERSE EVENT; QOL: QUALITY OF LIFE (3) SEE TEXT FOR INDIVIDUAL PROCEDURES AND EVALUATIONS & DEFINITION OF CLINICAL RECOVERY(PRIMARY EFFICACY)

**Fig 1. Study flow diagram showing study events and timelines.**

## Patient and public involvement

The patients were not involved in the preparation of the protocol or in carrying out study assessments and analyses. Numerous information bulletins on Ayurvedic remedies and research were posted by the Ministry of AYUSH from time to time [30].

## Selection, screening and eligibility, enrollment and management

During the first pandemic wave, all cases of COVID-19 including mild and suspected cases were to be admitted in Government accredited hospitals (COVID-19) [3]. Patients could directly access the hospital. They were first triaged by the general duty medical officer in the

emergency casualty department/outpatient. The diagnosis was confirmed by a standard real-time specific (SARS-CoV-2) reverse transcriptase polymerase chain reaction (RT-PCR) assay on a nasal and/or throat swab.

In the current study, the study investigators enrolled mild and moderate cases of COVID-19 which was classified as per Indian guidelines and clinical judgment [3]. Following hospital admission (study sites), the investigator explained the current study to those patients who expressed interest. Volunteer patients signed an informed consent form. The screening was completed within 48 hours of hospital admission as required by the protocol. Patients found eligible were quickly randomized.

Adult patients with a typical COVID-19 illness and a confirmed diagnosis were selected and after fulfilling the inclusion and exclusion criteria described in the protocol (enclosed, S2 File) were enrolled [24]. Patients with severe symptomatic COVID-19 were excluded if they satisfied any two of the following criteria (protocol) (i) respiratory distress at room ambiance (ii) Oxygen saturation (SpO2) at rest $\leq$ 93% (iii) known COVID-19 complication which may require oxygenation and/or critical care.

An Ayurvedic and a modern medicine physician in the study team supervised medical management on daily basis and along with nursing and paramedic personnel ensured drug compliance and reporting of all adverse events (AE). However, final decisions pertaining to the medical management of COVID-19 and recovery were taken by an independent hospital COVID-19 physician (blinded to treatment allocation). Patients and study personnel were aware of the specific study intervention allocation (open label).

## Randomization

Patients were randomized at each site to either of the two arms of a standard of care (SOC) or AYUSH 64 administered along with SOC (AYUSH plus) in a 1:1 ratio on a first come first serve basis. A central randomization schedule was prepared by the study biostatistician (SS) using standard statistical software (WINPEPI version 4.61 for MS Windows). Permuted block randomization was used in a group of 20 participants (strata of size 20) to ensure a number balance. The randomization schedule was provided online with restricted access to the site principal investigator.

## Standard of care (SOC)

SOC regimen was begun in-patient by an independent COVID-19 physician according to national and institutional recommendations [3]. However, the physician was permitted to use clinical judgement.

## Investigational drug

Each 500 mg tablet of AYUSH 64 contained aqueous extracts (100 mg each) of *Alstonia scholaris* (bark), *Picrorhiza kurroa* (rhizome), *Swertia chirata* (whole plant), and *Caesalpinia crista* (200 mg seed powder). The dose was two tablets of 500 mg each and taken twice daily with a glass of water soon after a meal, and this dosage remained fixed throughout the study. Patients assigned to the AYUSH 64 plus arm continued the drug following clinical recovery till the completion of the study period (12 weeks). AYUSH 64 was procured from Indian Medicines Pharmaceutical Corporation Limited (IMPCL), Uttarakhand, India under arrangements with CCRAS, New Delhi. The manufacturing facility was a certified ISO 9001 facility (2008) and followed good manufacturing practices' guidelines in the Ayurvedic Pharmacopoeia of India. Details of composition, quality standards, and features of chemistry, manufacturing, and controls are described in (Tables S3.1-S3.3, S3.1 Fig in S3 File).

## Outcome measures

The primary efficacy measure was (i) the mean duration (days) from baseline randomization to day one of clinical recovery (CR) and (ii) the proportion of patients showing clinical recovery, within a time framework of 28 days. Clinical recovery was accepted when all of the following criteria were met for at least 48 hours under the observation of the hospital COVID-19 physician (a) normal body temperature ($\leq 36.6°C$ axillae or $\leq 37.2°C$ oral) (b) absence of cough requiring regular medication (c)absence of breathlessness on routine daily self-care activities and respiratory rate less than 30 breaths per minute without supplemental oxygen (d) absence of any other symptom/sign attributed to COVID-19 illness and requiring continuous treatment (e) normal SpO2 by standard peripheral oximetry device (above 95 percent)(f) negative RT-PCR assay for SARS-CoV-2 from nasal and throat swab. The checklist of symptoms that were monitored daily for recovery was also guided by the WHO guideline [31].

There were several secondary efficacy measures pertaining to (i) timelines such as the mean duration from onset of symptoms to CR, mean duration from hospitalization to CR (ii) COVID-19-related blood assay biomarkers such as C-reactive protein (CRP), D-Dimer, Ferritin, interleukin-6. They are described in the enclosed protocol (S2 File).

## Procedures and measures

The assessment of general physical and mental health, psychosocial health, and quality of life (QOL) were carried out by using the standard World Health Organization QOL BREF questionnaire and a recently developed Health-Related-Behavior Habit and Fitness Questionnaire (HR-BHF CRD, Pune 2020 version) [32, S4 File]. Both questionnaires were self-reported and administered in the local language. While the WHO QOL recorded the response on a 5-point Likert categorical scale, the HR-BHF used a horizontal visual analog scale (VAS, 0–100 mm). Both the scales were anchored at either end to show the worst response or the best response, e.g., in the WHO-QOL questionnaire, category 1 was 'very poor' and category 5 was 'very good' for certain questions. In the case of HR-BHF, a VAS score of 0 indicated the worst response and a score of 100 indicated the best response for certain questions; the ascending order of better response was reversed for few questions to facilitate understanding by the patient (such as in case of 'anxiety' a VAS score of 0 meant absence of anxiety and score 100 meant maximum anxiety), and this was adjusted in the final score. A comprehensive description and scoring method including pre-study validation is provided in the Text Box S4.2, Text Box S4.3 in S4 File.

The WHO QOL-BREF had 27 questions classified into 4 domains- physical health (7–35), psychological health (6–30), social relationships (3–15), and environmental well-being (8–40); the range of score is shown in parenthesis.

HR-BHF contained nine questions pertaining to general health, anxiety, fatigue, energy level, bowel habits, stress, happiness, sleep, and appetite (food). Individual question score (0–100) and the total score (0–900) was used for analysis in the current study.

Standard procedures were used to classify, monitor, record, and assign causality, in the case of an adverse event (AE, System organ classification, clinical grade of severity, preferred terms for recording signs, symptoms, and diagnosis). Guidelines published by ICH-GCP, ICMR (India), MedDRA, and WHO were also followed [28, 33, 34]. All randomized participants were assessed for safety and tolerability (AE).

Clinical evaluation included a routine physical and systemic examination. Both the clinical and laboratory evaluation was carried out several pre-determined study time points which included randomization baseline and study completion (Fig 2). Routine laboratory measures included hematocrit, metabolic hepatic and renal profile, and urinalysis and were carried out in laboratories at each study site that was a-priori endorsed by the 'National Accreditation

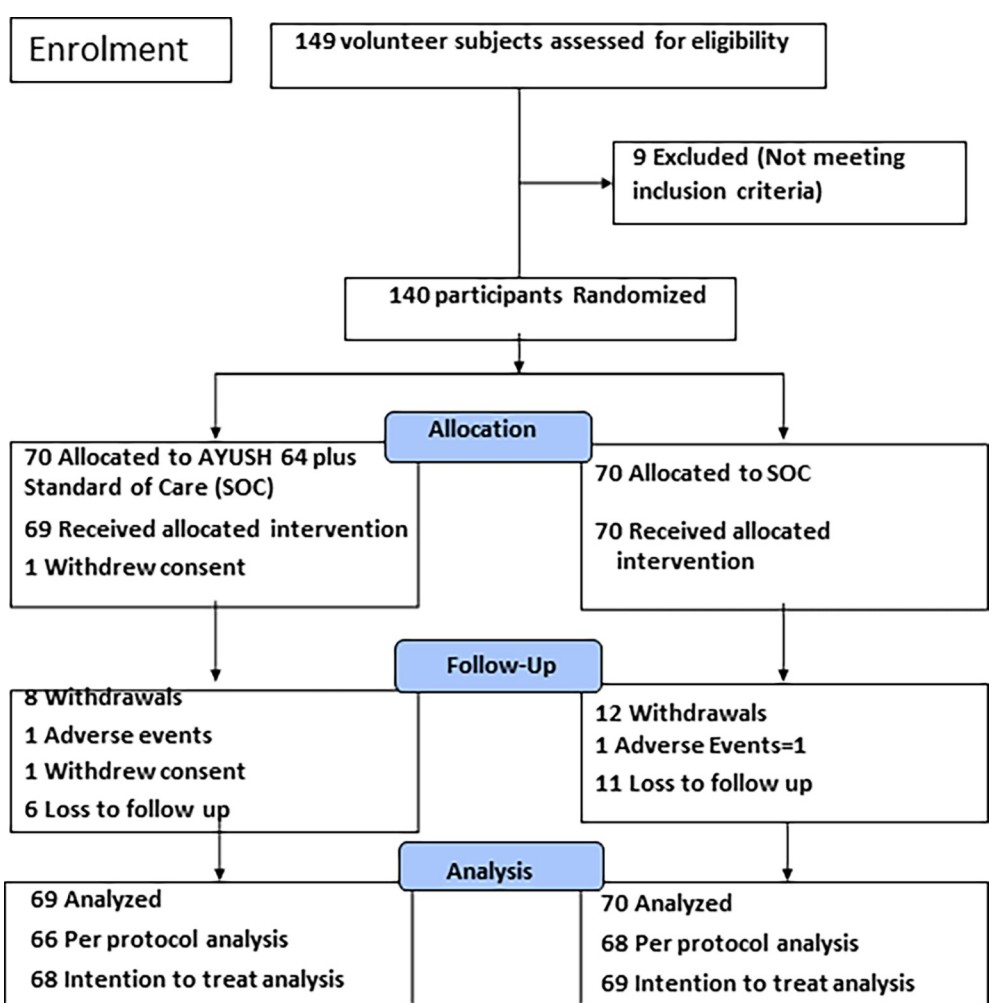

**Fig 2. Patient disposition and withdrawals: A randomized controlled study to evaluate the co-administration of AYUSH-64 with Standard of Care (SOC) in–mild-moderate symptomatic COVID-19 (CONSORT flow diagram).**

Board for Hospital and Health Care Providers' (NABH). Further, these laboratories were necessarily approved by the Indian Council of Medical Research (ICMR) to carry out real-time RT-PCR assays for diagnosis of SARS-CoV-2 infection as per the existent Indian Government policy. Laboratories followed standard ICMR recommendations for reagents, equipment and procedure, and quality checks [35].

Electrocardiography was recorded during screening, hospital discharge, and on study completion.

Standard skiagram of chest was carried out during screening, on clinical recovery/ hospital discharge, and during any follow-up visits if clinically indicated. The radiological evaluation described in the S5 File pertained to 86 participants at two study sites that provided digital skiagrams of satisfactory quality according to an independent radiologist. Due to hospital priorities, participants at one site were only screened if they had persistent respiratory complaints using a mobile X-Ray unit, and skiagrams were not printed; 6 participants were reported with mild abnormalities and none had severe disease. The latter data were not included in this report. For the current radiographic analysis, all the available digital skiagrams were centrally reassessed by an independent radiologist who was blinded to the allocation of study treatment.

Data were collected on a daily basis during the hospitalization phase. Subsequently, after hospital discharge, it was collected during the predetermined follow-up schedule (4, 8, and 12 weeks) as shown in Fig 2. Participants were counseled to contact the study physicians at any time during the follow-up if they showed any fresh symptoms or worsened or suspected a drug-related side-effect. Patients were also provided with a specially designed software program for mobile application called 'COVID KAVACH' for daily monitoring during the follow up. The latter was used by the patient to record any AE or study-related problem which was electronically communicated to the site investigator and the study coordinator (AC) [36].

Data were recorded in the study case report forms at the point of care and later entered into a central electronic database using unique participant ID and study treatment code. The database was handled by pre-designated study team paramedics and locked after validation for any errors by the designated in-charge from CRO. Prior to the data analysis, a backup copy was provided to the sponsor by the CRO. Th database was unlocked and decoded by the study biostatisticians (SS, ST) prior to statistical analysis.

## Study withdrawal

Patients worsening clinically and likely to require prolonged oxygen and/or intensive care were withdrawn from the study and transferred to intensive care for further management. All patients were comprehensively evaluated at the time of withdrawal. The site personnel continued to contact withdrawn patients for further disease progress and recovery and for the occurrence of any AE till such time the study was completed. The latter data were not included in the current report.

## Statistical analysis

There was no prior data to use for the formal calculation of a sample size. However, we considered relevant clues for a probable medium effect size which recommended 64 participants per group (type-I error = 0.05, power = 80%, Table 2) and was published in a classic reference [37]. Finally, after discussion with the study group experts, the principal investigator and coordinator (AC) and the chief biostatistician (SS) finalized a convenience (non-probabilistic) sample of 140 participants. This was considered adequate to address the clinical research questions. Other factors like study logistics [mainly available time & resources including manpower] and restrictions imposed by the pandemic were also considered.

The study data was entered by the designated personnel at each study site into a central database and supervised by the CRO. Data were summarized using central tendencies (mean, median), range, standard deviation, and 95% confidence interval (95% CI).

Statistical tests were carried out to compare the two treatment groups as per the distribution (normality) and the type & level of measurement of the variable under consideration (like Student's 't-test, Mann-Whitney non-parametric test, and Chi-square test) The result of statistical analysis was considered significant at $P < 0.05$ (two-sided). Both intent-to-treat (ITT) and per-protocol/completer (PP) analyses were performed for the primary efficacy analysis and some secondary measures. The ITT included all subjects who completed the randomized treatment observation till clinical recovery. The PP analysis included all subjects from the latter who were randomized within 48 hours of hospital admission and strictly adhered to other protocol requirements.

Participants completing the study intervention as per protocol were considered as qualified for the primary efficacy analysis using parametric (Student's 't-test) and non-parametric tests (Mann-Whitney statistic). Categorical outcomes such as AE were compared using Chi-square statistic. The primary efficacy measure was also analysed for the total number of participants at

each study site and in the study. Similarly, the two study groups were also compared for several secondary efficacy measures (timelines, laboratory assays, Quality of life measures) at several time points as per protocol.

A general mixed-effect linear regression model was also carried out with 'study site' as a random effect and 'group intervention ' as a fixed effect for the primary efficacy measure. In the latter case, 'Time from Randomization to Clinical Recovery' (primary efficacy) was the dependent variable.

Standard statistical software programs were used (GraphPad InStat Version 3.6, BMDP, IBM SPSS Version 20, and Confidence Interval Analysis, BMJ Group, London, 2003). The study arm of 'AYUSH 64 plus SOC' is referred to as 'AYUSH plus ' and 'SOC alone' is referred to as 'SOC' in the current paper.

## Results

A total of 140 participants were randomized with 70 participants in each of the two study arms- AYUSH plus and SOC (Fig 2).

### Withdrawals

Three participants were withdrawn during the randomization phase -one participant withdrew consent following randomization, one (AYUSH Plus) developed neuropathy (Guillain Barre syndrome) and one (SOC) developed severe pneumonia with respiratory distress. 137 patients completed the randomized treatment phase. A total of 20 (14.3%) participants withdrew (12 in the SOC group and 8 in the AYUSH Plus group) from the study. Seventeen patients did not wish to continue following complete recovery and hospital discharge. The latter did not report any AE during an informal follow up. None of the withdrawals were due to a drug related AE. There were no deaths in the study.

### Randomization baseline

Both the study groups were well matched for several demographic, clinical, COVID related timelines, SOC drugs and laboratory variables as shown in Table 1. 80% participants were clinically classified as mild COVID-19 at the time of enrolment and were mostly men in the age range 30–55 years. Several had comorbid disorders- hypertension, known diabetes, or first-time hyperglycemia (fasting blood sugar > 120 mg/dl).

There were no significant differences between the two study groups for COVID related timelines such as' onset of symptom to hospital admission' (-1.34 to 1.72),' hospital admission to randomization' (-0.17 to 0.39), and 'symptom onset to randomization' (-1.08 to 1.98); 95% CI of the difference (days) between means is shown in parenthesis (Table 1). Site-specific data for selected timelines and SOC drugs, including those related to RT-PCR assay, are shown in (Table S5.2 in S5 File).

A list of SOC drugs is shown in S5 File [R]. Most of the patients were treated with symptomatic drugs, vitamins and minerals. Several patients also received Hydroxychloroquine, or Ivermectin with Azithromycin (Table 1). At one site, anti-coagulants were empirically administered to patients with moderate COVID-19 and or with important risk factors (COVID-19). Except for one patient, none of the trial participants were treated with steroids. Parenteral Dexamethasone was administered to only one patient with progressive respiratory distress who was withdrawn from the study. Importantly, both the intervention study groups were well matched for the use of various drugs (Table 1).

62.8% of the participants showed radiographic abnormalities in the chest which were consistent with COVID-19 and classified as mild or moderate by the radiologist (See, Table S5.6 in S5 File).

**Table 1. Randomization baseline data on demographic, clinical, COVID related timelines, laboratory variables, and Standard of Care (SOC) drugs in the study groups.**

| VARIABLES | AYUSH Plus (n = 69) | SOC (n = 70) | P-value* |
|---|---|---|---|
| **Clinical** | | | |
| Age (years) Mean ± SD | 42.87 ± 12.6 | 42.7 ± 12.0 | 0.93 |
| Male–number (%) | 54 (77.14%) | 58 (82.85%) | 0.52 |
| Body Weight (kg) Mean ± SD | 69.34 ±10.3 | 68.38 ±12.1 | 0.61 |
| BMI (kg/m$^2$) Mean ± SD | 24.86 ±3.4 | 24.53 ±3.7 | 0.65 |
| Symptom onset to randomization (days), mean ± SD | 7.61 ±4.8 | 7.83 ± 4.5 | 0.51 |
| Symptom onset to Hospitalization, mean ± SD | 6.4 ± 4.64 | 6.5 ± 4.47 | 0.75 |
| Hospitalization to Randomization, mean ± SD | 1.4 ± 0.8 | 1.5 ± 0.9 | 0.55 |
| Mild clinical disease number (%) | 56 (80) | 58 (82.9) | 0.82 |
| Moderate clinical disease number (%) | 14 (20) | 12 (17.1) | 0.82 |
| Hypertension number (%) | 17 (24.29) | 10 (14.29) | 0.19 |
| Diabetes mellitus-number (%) | 14 (20) | 06 (8.57) | 0.09 |
| Undiagnosed hyperglycemia-number (%) | 9 (12.85) | 14 (20) | 0.36 |
| Blood sugar level mg/dl, mean ± SD | 112.50 ± 37.5 | 114.17 ± 35.2 | 0.74 |
| ESR mm fall 1$^{st}$ hour, mean ± SD | 50.2 ± 38.0 | 46.9 ± 37.4 | 0.79 |
| Blood hemoglobin gm/dl, mean ± SD | 13.6 ± 1.42 | 13.8 ±1.62 | 0.51 |
| Total leucocyte count/cu mm, mean ± SD | 5920.7 ± 2008 | 6828.3 ± 2085 | 0.02 |
| Total Lymphocyte count/cu mm, mean ± SD | 32.31 ± 9.1 | 31.07 ± 09.6 | 0.34 |
| **Symptoms at baseline-number of subjects (percent)** | | | |
| Fever | 53 (75.71%) | 45 (64.28%) | 0.10 |
| Sore throat | 46 (65.71%) | 53 (75.71%) | 0.24 |
| Cough | 54 (77.14%) | 54 (77.14%) | 0.87 |
| Dyspnea | 24 (34.28%) | 25 (35.71%) | 0.90 |
| Myalgia | 48 (68.57%) | 54 (77.14%) | 0.31 |
| Headache | 37 (52.85%) | 32 (45.71%) | 0.35 |
| Diarrhea | 11 (15.71%) | 12 (17.14%) | 0.85 |
| Ageusia | 19 (27.14%) | 19 (27.14%) | 0.96 |
| Anosmia | 13 (18.57%) | 14 (20%) | 0.86 |
| **Drugs administered-number of subjects (percent)** | | | |
| Tab Azithromycin | 48 (70%) | 49 (70%) | 0.95 |
| Tab Doxycycline | 1(2%) | 0 | 0.31 |
| Tab HCQS | 29 (42%) | 24(34%) | 0.35 |
| Tab Zinc | 48(70%) | 42 (60%) | 0.24 |
| Tab Vitamin C | 69 (100%) | 69(99%) | 0.32 |
| Tab Vitamin D 3 | 15(22%) | 18(26%) | 0.58 |
| Tab Pantoprazole | 66(96%) | 65(93%) | 0.73 |
| Tab Paracetamol | 59 (86%) | 55(79%) | 0.28 |
| Tab Cetirizine | 13(19%) | 15(21%) | 0.70 |
| Tab Ivermectin | 3 (4%) | 2 (2.9%) | 0.99 |
| Anti-coagulant | 13(18.8%) | 13(19%) | 0.97 |
| Oxygen intermittent (< 2 liters/min) | 9(13%) | 6 (9%) | 0.39 |

*Statistically significant (P<0.05)

NS Not statistically significant (P> = 0.05)

a. Student's 't test (normative data) or Chi-Square statistic (categorical data)

b. Note- n: number of study participants; SD: standard deviation; AYUSH plus: AYUSH 64 + SOC; BMI: body mass index; ESR: erythrocyte sedimentation rate (Wintrobe method); Other comorbid disorders: hyperlipidemia (2), cardiac disorder (1), chronic lung disease (2), thyroid disorder (5), Allergic rhinitis (3), number of participants in parenthesis; See text for details

## Efficacy

134 participants qualified for the primary efficacy analysis and the results on per protocol analysis are shown in Table 2; three study participants were disqualified because of delay in randomization. The mean duration (days) for clinical recovery (primary efficacy) from the randomization baseline was significantly superior in the AYUSH plus (6.45, 95% CI 5.88 to 7.01 days) as compared to SOC (8.26, 95% CI 7.20 to 9.31 days); difference between means -1.81 (95% CI—3.02 to—0.59 days) (Table 2). Significant improvement was also observed at each of the study sites.

At this stage, we did not intend performing statistical analysis for predictor of response to treatment. However, the results of a general linear mixed effect model using the primary efficacy data as a dependent variable are shown in Table S5.4 in S5 File and indicate that the outcome remained significantly different between the two study groups even after controlling/removing the effect of 'study site'.

In an intention to treat analysis (137 participants), the mean duration (days) for clinical recovery (primary efficacy) from the randomization baseline was significantly superior in the AYUSH plus (6.42, 95% CI 5.99 to 7.59) as compared to SOC (8.33, 95% CI 7.02 to 9.87 days); difference between means was -1.90 (95% CI—3.11 to—0.70 days) (Table S5.5 in S5 File). This improvement was also observed at each of the study site.

A higher proportion of patients in the AYUSH plus (69.75%) showed complete recovery as compared to SOC (52.9% patients) during the first week following randomization (P = 0.046, Chi-square statistic).

**Table 2. Primary efficacy measure (Randomization to clinical recovery) and selected timeline in the two study groups (n = 134) in per protocol analysis; mean (days)± standard deviation.**

| Time line (days) | Mumbai (n = 57) | | Nagpur (n = 29) | | Lucknow (n = 48) | | Total Study (n = 134) | |
|---|---|---|---|---|---|---|---|---|
| | AYUSH Plus (n = 28) | SOC (n = 29) | AYUSH Plus (n = 15) | SOC (n = 14) | AYUSH Plus (n = 23) | SOC (n = 25) | AYUSH Plus (n = 66) | SOC (n = 68) |
| Randomization to Clinical Recovery (R-CR) | | | | | | | | |
| Mean ± SD (R-CR) | 6.75± 2.14 | 8.45± 3.75 | 8.80± 1.01 | 11.21± 5.03 | 4.57± 1.56 | 6.40± 4.03 | 6.45± 2.35 | 8.26± 4.44 |
| 95% CI of Difference between Means (R-CR) | -3.32 to -0.07 | | - 5.37 to 0.54 | | -3.61 to—0.06 | | - 3.02 to—0.59 | |
| On comparison of R-CR between two intervention groups at each study site and study cohort * | P = 0.0410 ('t' test),0.077 (M-W) | | P = 0.079 ('t' test), 0.013 (M-W) | | P = 0.046 ('t' test),0.121 (M-W) | | P = 0.003 ('t' test), 0.015 (M-W) | |
| Onset symptom to Clinical Recovery (S-CR) | | | | | | | | |
| Mean ± SD (S-CR) | 15.29 ± 6.15 | 16.45 ±5.92 | 11.07± 1.58 | 14.50 ± 5.60 | 11.52 ± 3.26 | 13.48 ± 6.00 | 13.02 ± 4.87 | 14.96 ± 5.95 |
| 95% CI of Difference between Means (S-CR) | -4.37 to 2.05 | | -6.76 to -0.11 | | -4.75 to 0.84 | | -3.79 to -0.08 | |
| On comparison of S-CR between two intervention groups at each study site and study cohort * | P = 0.470 ('t' test), 0.592 (M-W) | | P = 0.031('t' test), 0.016 (M-W) | | P = 0.172 ('t' test), 0.525 (M-W) | | P = 0.041('t' test), 0.066 (M-W) | |

*Statistically significant (P<0.05)

NS Not statistically significant (P> = 0.05)

a. two independent samples 't' test and M-W (Mann-Whitney statistic

b. Note, AYUSH plus: AYUSH 64 + Standard of Care (SOC); 134 patients qualified for primary efficacy analysis; Clinical recovery was essentially absence of COVID 19 symptoms for two successive days (with negative RT-PCR assay); See Text for detail

The earlier recovery in the AYUSH plus group was also observed for 'time to clinical recovery from the onset of symptom' and this was marginally significant as compared to SOC (Table 2).

There was a significant reduction in serum biomarkers of COVID-19 in each of the study groups but the difference between the groups was not significant (Table 3).

Eight participants (18.6%) in the AYUSH plus and 13 participants (30.2%) in the SOC group showed radiological features of definite COVID-19 pneumonia at the time of randomization baseline (See Table S5.6 in S5 File). A higher number of participants (84.6%) in the SOC showed incomplete resolution as compared to AYUSH plus SOC (62.5%) at the time of clinical recovery (Table S5.6 in S5 File). None of the patients with COVID-19 radiographic abnormalities during the initial hospitalization complained of fever, persistent cough, or continuous breathlessness on follow up till completion of the study. There were no clinically suspected post-COVID lung complications and the skiagrams of several patients were reported normal.

In comparison to SOC, AYUSH Plus showed significant improvement in several domains (physical health, psychological health, social relationship, and environmental well-being) in the WHO QOL BREF and the total HR-BHF score at the time of clinical recovery and during follow-up (Table 4). It was notable that the AYUSH Plus also showed significant improvement in several individual item scores (fatigue, stress, anxiety, appetite, and happiness) in the HR-BHF questionnaire as compared to SOC (See Table S4.1 in S4 File).

## Safety and related issues

28 patients in the AYUSH Plus and 29 patients in the SOC group reported adverse events: there were no statistically significant differences (Table 5). Each of the study group reported 48 AE. Additional data on AE is shown in Table S6.1 in S6 File.

**Table 3. Selected biomarkers related to COVID-19 in the two study groups (n = 139); mean (days) ± standard deviation.**

| Variable | Study groups | Baseline | On discharge* | Week 12* |
|---|---|---|---|---|
| Lactose dehydrogenase (LDH) | AYUSH plus | 403.6 ± 131.6 | 338.9 ± 109.7 | 318.7 ± 109.6 |
| | SOC | 446.7 ± 206.5 | 363.8 ± 115.2 | 381.9 ± 164.8 |
| LDH Reference Range: 225–480 U/L | | | | |
| Ferritin | AYUSH plus | 337.8 ± 280.3 | 257.7 ± 226.1 | 84.4 ± 70.2 |
| | SOC | 337.4 ± 278 | 201.8 ± 206 | 92.5 ± 89.3 |
| Ferritin Reference Range: Male 30–350 ng/ml; Female 20–250 ng/ml | | | | |
| Procalcitonin | AYUSH plus | 0.1 ± 0.1 | 0.1 ± 0.1 | 0.1 ± 0.2 |
| | SOC | 0.1 ± 0.04 | 0.1 ± 0.2 | 0.1 ± 0.2 |
| Prolactin Reference Range: <0.2 ng/ml | | | | |
| C-reactive protein | AYUSH plus | 20.83 ± 27.55 | 10.3 ± 19.1 | 6.3 ± 6.5 |
| | SOC | 25.5 ± 35.3 | 10.7 ± 12.5 | 6.39 ± 8.98 |
| C-Reactive Protein Reference Range <3 mg/L | | | | |
| D-Dimer | AYUSH plus | 462.5 ± 439.9 | 334 ± 224.9 | 297.3 ± 277.6 |
| | SOC | 523.2 ± 672.8 | 345.3 ± 324.2 | 317.9 ± 418.4 |
| D-Dimer Reference Range:0–400 ng/ml | | | | |
| Interleukin-6 | AYUSH plus | 30.6 ± 46.0 | 7.7 ± 12.2 | 8.5 ± 22.1 |
| | SOC | 32.6 ± 42.2 | 8.5 ± 15.8 | 7.4 ± 10.3 |
| Interleukin-6 Reference Range: Up to 7 pg/ml | | | | |

*Statistically significant (P<0.05, Mann Whitney statistic) change from baseline within the study group for all variables except Pro-calcitonin

**NS Not statistically significant (P> = 0.05)

Note, Abbreviation: AYUSH plus: AYUSH 64 + Standard of Care (SOC); n = number of participants; See text for detail

**Table 4. Quality of life questionnaires and health scores (HR-BHF and WHO QOL Bref) in the two study groups (n = 139).**

| Variable | Baseline (n = 139) | Discharge (n = 137) | Week 4 (n = 129) | Week 8 (n = 127) | Week 12 (n = 120) |
|---|---|---|---|---|---|
| Health-Related- Behavior, Habit and Fitness (HR-BHF) questionnaire: combined score | | | | | |
| AYUSH plus | 500.1 ± 89.9 | 667.4 ± 85.7* | 690.7 ± 111* | 721.6 ± 105.5* | 748.1 ± 114.5** |
| SOC | 493.4 ± 81 | 637.5 ± 81.1 | 650.6 ± 100.6 | 677.7 ± 89.9 | 682.4 ± 90.9 |
| WHO BREF Quality of life (QOL) Domain I (Physical health) | | | | | |
| AYUSH plus | 24.6 ± 4.1 | 28.8 ± 2.2* | 28.9 ± 2.34 | 30.0 ± 2.15 | 30.2 ± 2.07 |
| SOC | 23.05 ± 4.42 | 27.8 ± 2.82 | 28.6 ± 2.7 | 29.3 ± 1.8 | 29.4 ± 2.1 |
| WHO BREF Quality of life (QOL) Domain II (Psychological health) | | | | | |
| AYUSH plus | 20.81 ± 3.67 | 23.48 ± 2.28 | 24.2 ± 1.51 | 24.6 ± 1.70* | 24.7 ± 1.88 |
| SOC | 20.1 ± 4.04 | 23.2 ± 2.29 | 23.4 ± 2.21 | 23.9 ± 1.57 | 24.1 ± 1.88 |
| WHO BREF Quality of life (QOL) Domain III (Social health) | | | | | |
| AYUSH plus | 10.21 ± 2.03 | 11.34 ± 1.33 | 11.98 ± 1.02 | 12.05 ± 1.19* | 12.30 ± 1.22** |
| SOC | 10.26 ± 2.21 | 11.52 ± 1.29 | 11.74 ± 1.35 | 11.55 ± 1.48 | 11.62 ± 1.25 |
| WHO BREF Quality of life (QOL) Domain IV (Environmental health) | | | | | |
| AYUSH plus | 27.27 ± 4.68 | 30.81 ± 2.30 | 31.74 ± 2.40 | 32.32 ± 2.73 | 32.22 ± 2.52* |
| SOC | 26.66 ± 5.21 | 30.38 ± 2.45 | 30.90 ± 2.54 | 31.55 ± 2.06 | 31.43 ± 2.32 |

*Statistically significant (P<0.05, Mann Whitney statistic)

**Statistically highly significant (P<0.01, Man Whitney statistic)

NS Not statistically significant (P> = 0.05)

a HR-BHF total score and WHO BREF QOL domain score (physical, psychological social, and environmental health) were significantly better in AYUSH plus study group at several study time points

b. Note, See S4 File for methods and scoring of WHO QOL Bref and HR-BHF (Health Related-Behavior Health and Fitness) questionnaire; HR-BHF score range 0–900; WHO QOL Bref domain scores vary as shown in S 4 File but higher score generally meant better outcome; n: number of participants; AYUSH 64 plus: AYUSH 64 plus Standard of Care (SOC); See text for details

AE were generally mild in nature and pertained to episodic fever, myalgias, fatigue, occasional breathlessness, loss of taste and/or smell and were mostly reported during the post-hospitalization follow-up. Several AE were possibly symptoms of COVID-19 rather than due to any study drug. A probable or definite causality of AE with AYUSH 64 could not be confirmed in any of the study participants. However, based on a-priori knowledge and experience of the Ayurvedic physicians in the study, some of the gut-related AE, albeit mild, which were present in the AYUSH plus may have been due to AYUSH 64 medication. Most of the time, AE did not require any specific treatment. Three participants reported serious AE and all recovered without any complications. Moderate AE was treated symptomatically. Those suspected of severe AE were referred to a specialist for an opinion. Participants with naïve hyperglycemia and/ or dyslipidemia were managed by a specialist physician.

Clinically, none of the AE was related to a drug interaction.

Repeated routine laboratory assays remained within normal limits in the two arms and there were no significant differences between the treatment arms Table S7.1 in S7 File. Electrocardiography of all participants was reported normal at baseline, hospital discharge, and on study completion.

## Discussion

This randomized controlled multicentric study showed that a combination regimen of AYUSH 64, a standard Ayurveda drug, and SOC was significantly superior to SOC in the treatment of mild and moderate COVID-19. 140 eligible participants were randomized for study intervention and monitored under direct physician observation in an in-patient COVID

**Table 5.** Adverse events (AE) in the two study groups* (n = 139).

| Adverse Events | | AYUSH plus (n = 69) | | OC (n = 70) | |
|---|---|---|---|---|---|
| | | Participant | Events | Participant | Events |
| **Summary** | | | | | |
| Total | | 28 | 48 | 29 | 48 |
| Mild | | 14 | 27 | 14 | 33 |
| Moderate | | 12 | 19 | 13 | 14 |
| Severe | | 2 | 2 | 2 | 2 |
| Serious AE | | 1 | 1 | 2 | 2 |
| **Causality** | | | | | |
| Unrelated | | - | 45 | - | 47 |
| Unlikely | | - | 03 | - | 2 |
| **AE is classified according to System Organ Classification and Preferred Term/Diagnosis (according to the investigator)** | | | | | |
| Cardiac | | 1 | 1 | 0 | 0 |
| | Transient hypertension | 1 | 1 | 0 | 0 |
| Ear and labyrinth | | 0 | 0 | 1 | 1 |
| | Ear ache | 0 | 0 | 1 | 1 |
| GIT | | 10 | 10 | 5 | 5 |
| | Gastritis | 0 | 0 | 1 | 1 |
| | Dyspepsia | 1 | 1 | 1 | 1 |
| | Diarrhea | 4 | 4 | 1 | 1 |
| | Constipation | 2 | 2 | 1 | 1 |
| | Epigastric pain | 1 | 1 | 0 | 0 |
| | Vague pain | 0 | 0 | 1 | 1 |
| | Hyperacidity | 2 | 2 | 0 | 0 |
| Hepatic | | 1 | 1 | 0 | 0 |
| | Transaminitis | 1 | 1 | | |
| Infections & infestations | | 5 | 6 | 6 | 9 |
| | Episodic Fever | 5 | 5 | 2 | 2 |
| | Malaria | 0 | 0 | 2 | 2 |
| | Cellulitis | 0 | 0 | 1 | 1 |
| | Sore throat | 1 | 1 | 3 | 4 |
| Skin | | 2 | 2 | 1 | 1 |
| | Itch | 1 | 1 | 0 | 0 |
| | Eczema | 1 | 1 | 0 | 0 |
| | Non-specific | | | 1 | 1 |
| Respiratory | | 6 | 7 | 7 | 8 |
| | Cough | 1 | 1 | 2 | 2 |
| | Episodic Breathlessness | 5 | 6 | 4 | 5 |
| | Non-specific | 0 | 0 | 1 | 1 |
| Nervous system | | 3 | 3 | 0 | 0 |
| | Neuropathy | 1 | 1 | 0 | 0 |
| | Vertigo | 2 | 2 | 0 | 0 |
| Renal | | 0 | 0 | 1 | 1 |
| | Dysuria | 0 | 0 | 1 | 1 |
| Endocrine | | 6 | 6 | 6 | 6 |
| | Hyperglycemia | 6 | 6 | 6 | 6 |
| Investigation (Laboratory) | | 0 | 0 | 1 | 1 |
| | Hyperlipidemia | 0 | 0 | 1 | 1 |

(*Continued*)

**Table 5.** (Continued)

| Adverse Events | | AYUSH plus (n = 69) | | OC (n = 70) | |
|---|---|---|---|---|---|
| Others | | 12 | 13 | 12 | 15 |
| | Weakness | 2 | 2 | 5 | 6 |
| | Chills | 0 | 0 | 1 | 1 |
| | Myalgia | 5 | 6 | 6 | 8 |
| | Arthralgia | 2 | 2 | 0 | 0 |
| | Headache | 3 | 3 | 0 | 0 |

(1) Abbreviations: AYUSH Plus: AYUSH 64 plus Standard of Care (SOC); n: number of participants; GIT: gastrointestinal

(2) Clinical grading as per WHO classification

(3) Causality in the AYUSH plus pertained to AYUSH 64 drug while causality in the SOC arm was not specified to any particular drug

(4) Transaminitis: raised serum glutamic oxalacetate and or pyruvate

(5) No AE recorded for Disorders of blood and lymphatic, immune system, metabolism and nutrition, psychiatric, reproductive system and breast, eye, vascular system, congenital familial and genetic, injury poisoning and procedural complications, and surgical and medical procedures

(6)See Text and S6 File for further detail

hospital setting. The 95% confidence interval of the difference in the mean duration (days) of clinical recovery (a-priori definition) from randomization baseline was—3.02 to—0.59 days (Table 2) as per the protocol analysis and -3.11 to -0.71 as per the intention-to-treat analysis Table S5.5 in S5 File in favor of the AYUSH plus intervention. The latter was also shown at each study site. A significantly higher proportion of AYUSH plus participants (69.7% versus 51.7%) achieved clinical recovery within the first week after randomization. AYUSH Plus also showed substantial, and often significant, improvement in several secondary efficacy and quality of life measures (Tables 2, 4).

AYUSH 64 was well tolerated and found safe over 12 weeks of use in the dosage prescribed in the current study. There were no differences in the AE between the two study groups. AE were generally mild, and none caused the withdrawal of participants. Only 3 serious AE were reported (2 in SOC). 20 participants withdrew from the study and mostly after clinical recovery as per the personal preference not to continue in the study. There were no deaths in the study.

Though enrolled with mild and moderate COVID-19, several participants in the current trial also suffered from chronic co-morbid disorders (Table 1) that have been reported to be risk factors for severe, progressive, and fatal disease [2–4, 6, 7]. Several naïve participants showed hyperglycemia on enrolment (Table 1) which has been reported to complicate recovery [38]. One participant in the AYUSH Plus developed Guillain Barre Syndrome which has been uncommonly reported as a COVID-19 neurological complication [39]. Over 60% of participants showed radiographic abnormalities consistent with COVID-19. Following recovery, none of the participants complained of persistent respiratory symptoms or were diagnosed with pulmonary fibrosis during the prolonged follow up. It is prudent to add that the current study protocol did not recommend CT scan of the chest for the diagnosis of an asymptomatic pulmonary sequel. Respiratory disorders including pulmonary fibrosis, and which are often asymptomatic, have been reported as an important COVID-19 complication [40].

COVID-19 is a dreadful disease with a huge burden of psychosocial disorders [41]. A meta-analysis from India reported several psychological comorbidities ranging from 26% (anxiety and depression) to 40% (poor sleep quality) of study participants [42]. In the current study, WHO QOL Bref and HR-BHF questionnaires were used. The significantly superior improvement in both the physical (including fatigue) and mental health (such as reduction in anxiety,

and stress) shown in the AYUSH Plus was clinically important and needs to be emphasized (Table S4.1 in S4 File). Several Ayurvedic medicines including the herbal ingredients of AYUSH 64 are reported to improve mental health [15, 17, 18, 20]. Several other QOL measures also showed a better improvement in the AYUSH Plus group (Table 4). In the passing, we wish to add that our study participants found it easier to answer visual analog scale-based questions in the HR-BHF questionnaire as compared to the somewhat cumbersome but popular WHO QOL BREF questionnaire (S4 File) [32].

More participants in the SOC arm showed definite radiographic pneumonitis (30% versus 19%) and failed resolution of radiographic abnormalities (85% versus 63%) at the time of clinical recovery/hospital discharge. One patient with mild radiographic disease in the SOC arm developed acute onset of progressive respiratory distress and required oxygenation and intensive care for recovery. All of this may suggest a more serious form of disease in the SOC arm, but this does not seem to be the case as shown by several other clinical variables, serum biomarkers, and overall clinical progress and response to standard of care treatment (Tables 1, 3 and S5 File). No uniform protocol was followed for radiological evaluation in the current study. Radiographic abnormalities often persist beyond clinical recovery and take a longer time for resolution, and often do not conform to the clinical severity of symptoms or disease [40]. Also, there is insufficient data on a prospective evaluation of radiographic abnormalities shown by conventional skiagrams in COVID-19 [40].

This study was exploratory in design and carried out during the first year of the pandemic. There were several concerns while the preparing the protocol. The pandemic and the stringent lock down imposed unique challenges for enrollment, physical and other examination, and monitoring of study participants. Our overall experience was consistent with that described recently in a report on drug trials in COVID-19 [43]. People were intensely scared and reluctant to participate. All treatment was mostly empirical and based on repurposed drugs [1–3]. There was no uniform protocol for standard care [3]. There were ethical issues with the use of placebo and blind study design.

## Strengths and limitations

The study data was captured using a pragmatic protocol. Presuming a modest effect size, a sample size of 128 subjects was suggested [37]. However, as there was no prior data to guide the latter assumption (see section above on statistical analyses), a convenience sample of 140 participants was agreed by the study experts. The concern of a selection bias due to a convenience sample size and an open label design seemed to have been nicely addressed by the randomization process and the study drug administration under direct medical observation during the hospitalization phase. Encouragingly, the two intervention groups were found to be well matched for several variables at randomization baseline (Table 1). The daily diligent clinical observation in the hospital ensured good compliance to the study intervention and robust efficacy data. The latter was also crucial for capturing AE and any obvious drug interaction. The tolerability and safety profile of AYUSH 64 was good and reassuring.

The confirmation of 'Clinical Recovery' (CR) in the current study may seem to be unduly subjective but was based on a pre-determined set of stringent criteria which included clinical and investigation measures (normal peripheral oximetry and a negative standard RT-PCR assay). Importantly, 48 hours of observation was mandatory to declare the resolution of symptoms and the total assessment was performed in a blinded manner. To the best of our knowledge, we did not find use of a similar set of criteria in any other interventional drug trial in COVID-19 during a search for relevant literature in the current study [44–46]. We did not measure 'viral load' as was performed in most of the drug trials [44]. The viral load may not

correlate with symptoms in mild and moderate disease, and during recovery [45]. It is notable that none of the study participants reported any clinical post COVID complication and actually improved their general physical and mental health during the follow-up period (Table 4).

There were other limitations that may have influenced the study outcome. Enrolment of subjects with early illness was a complex issue as was observed in several other COVID drug trials [43–45]. The delay was about a week from the onset of symptoms (Table 1 and S5 File). Investigations were not carried out in a central lab due to difficult logistics, but all study site laboratories were accredited (national standards) and compelled to strictly adhere to the guidelines on molecular testing for SARS-CoV-2 and quality control [35].

We were concerned about the surreptitious use of Ayurvedic drugs in the current drug trial. Ayurveda drugs and other popular traditional home remedies were extensively used in India during the pandemic and AYUSH 64 was available in the market [19, 20, 23]. None of the study participants admittedly used Ayurvedic drugs prior to hospital admission. It is unlikely that any medicine other than that permitted in the current study was taken by the participants during the in-patient treatment phase. Patients were counseled regarding medication by the study physician at the time of hospital discharge. Only the AYUSH plus participants were to continue AYUSH 64 drug till study completion. A special mobile software application (see methods) was used to maintain regular contact with the study participants. It is noteworthy, that several participants who continued AYUSH 64 showed better improvement in physical and mental health during the prolonged follow-up (Table 4).

The current study dealt with mild and moderate COVID-19 and no extrapolation of the outcome can be made to progressive and or severe disease. By the last quarter of 2020, the management of mild and moderate COVID-19 was rapidly shifted to a domiciliary or a quarantine facility in India [47]. Though the current trial participants were treated in a hospital setting (current study), it seemed fair to recommend the use of AYUSH 64 in a domiciliary or a quarantine setting under appropriate medical supervision [48].

## Mechanism of action

The human host, and not the microbe, is the therapeutic focus in Ayurveda while treating infections. The primary objective is to strengthen immunity. Ayurvedic physicians use a holistic approach to treat and heal which includes assessment of the individual constitution (called Prakruti and Doshas in Ayurveda) and several lifestyle changes [15, 21]. The pharmacological and therapeutic properties and experimental evidence (non-clinical) of AYUSH 64 and its ingredient medicinal plant extracts were recently published [18]. Some of the purported therapeutic properties were antipyretic, anti-infective, anti-inflammatory, anti-allergic, and immunomodulatory (called Rasayana in Ayurveda) [18, 26, 49].

Several experimental studies (animal, cell culture, and in-vitro model) of individual plant extract ingredients of AYUSH 64 have provided a wide array of evidence to explain the reduction in inflammation and modulation of immune response (anti-oxidant effect, increased phagocytosis and altered inflammatory pathways- Nuclear Factor Kappa B, p 65), direct inhibition of pro-inflammatory biochemical mediators and cytokines (prostaglandins, tumour necrosis factor alfa, Interleukin (IL)-1 beta, IL-6, and IL- 8), and suppression of inflammatory and allergic response in airways (cellular and cytokines) [18, 26, 49–53]. Interestingly, several inhibitory effects were also shown against viral protein R (HELA cells and plasmids) and some specific viruses (such as Herpes Simplex Type I, Coxsackie B2, Adenovirus, Poliovirus, and Chikungunya) [25, 50–53]. In a more recent in-silico molecular docking study, several ingredients of AYUSH 64 (and especially Akuammicine N-Oxide from Alstonia scholaris) showed good binding with the main protease enzyme of the SARS-CoV-2 [54]. AYUSH 64 showed

uncomplicated recovery with lesser requirement of symptomatic drugs and good safety when administered along with standard symptomatic treatment to 38 patients suffering from influenza-like illness in a prospective uncontrolled study of about one week duration [26].

## Other selected studies

Several Ayurvedic drug trials in COVID-19 were registered during the pandemic and the results of few published studies were encouraging [55, 56]. A uniform wholesome Ayurvedic regimen showed a reduction in the viral load in asymptomatic and early COVID-19 patients in a randomized placebo-controlled drug trial study but did not provide sufficient clinical data [57].

A meta-analysis of 18 randomized controlled drug trials showed the clinical benefit of co-administration of Chinese Herbal Medicine (CHM) with conventional Western Medicine (WM) in treating COVID-19; two trials showed that CHM plus WM significantly reduced hospital stay (95% CI of the mean difference -3.28 to -0.70 days) [14]. There were several methodological differences between the latter and the current study but intriguingly, the outcome of a mean reduction in hospital stay was almost similar. Superior cure rates and amelioration of individual symptoms were reported in a more recent systematic review and meta-analysis of controlled drug trials which used a combination of CHM and WM in mild to moderate COVID-19 [58].

Of late, monoclonal antibodies (MAB) are at the forefront of treating COVID-19 although they are presently contraindicated in severe and progressive disease [59]. However, MAB are specific for a particular SARS-CoV-2 variant and are recommended for use in subjects with early disease and risk factors (COVID-19). However, several issues connected with cost and logistics are hurdles in their clinical use in the Indian context [45]. Oral drugs like AYUSH 64 hold a greater appeal.

Despite extensive clinical use during the pandemic, CAM therapies such as Ayurveda and Traditional Chinese medicine seemed under-reported. [14, 20, 58]. A recent report based on a large cross-sectional survey of in-patients treated for COVID-19 described the use of repurposed and adjuvant modern medicines but failed to make mention of herbal drugs or other CAM therapies [60].

## Study implications and dissemination of results

In view of the lack of evidence for effective and safe drug therapy in COVID-19, several potential Ayurveda drugs and CAM were selected for repurpose and accelerated research and development [1, 2, 8, 9, 42]. Overall, the data from mild and moderate COVID-19 drug trials was sparse [43–45]. The current drug- trial of AYUSH 64 ought to be viewed from this perspective. In our experience, the success of the current study provided a substantial boost to the ongoing research efforts in Ayurveda and CAM. We believe that it will also encourage an integrative medicinal approach to treating difficult diseases like COVID-19.

It is prudent to add that several medicinal plant ingredients of AYUSH 64 have been used to promote health and treat diverse medical disorders for several centuries by physicians and traditional healers in India, China, Southeast Asia, Europe, and North America [18, 26, 49–51].

Study participants were informed about the current study results telephonically and/or through small virtual meetings by investigators and coordinators. A widely circulated Government press release in May 2021 announced the core study results and promoted the use of AYUSH 64 in COVID-19 [58]. A national education program was launched by the Ministry of AYUSH to disseminate information about AYUSH 64 and other drugs [61]. Along with the

latter, a nationwide distribution campaign (AYUSH 64) was also launched [61]. Simultaneously, the Ministry of AYUSH launched an evidence-based management protocol for Ayurveda and Yoga for the management of COVID-19 which contained a reference to the current study [48].

## Future research

AYUSH 64 ought to be further evaluated for the treatment of mild and moderate COVID-19, both as mono and a combination therapy (modern medicine), in a phase III drug trial. Studies should also evaluate the potential of AYUSH 64 to block progression of COVID-19 to severe disease and reduce post-COVID-19 complications. Experimental evidence is required to validate its anti-viral and other health benefits.

## Conclusion

AYUSH 64 (a standardized polyherbal Ayurveda drug) was shown to be a significantly effective and safe adjunct in the treatment of mild and moderate COVID-19 in a prospective, randomized controlled drug trial. Open-label study design and other limitations necessitate judicious interpretation and extrapolation of the current study data and outcome. AYUSH 64 hastened clinical recovery, reduced hospitalization period, and showed early persistent health benefits with minimal/ absent drug-related side effects.

## Supporting information

**S1 File. CONSORT check list.**
(DOC)

**S2 File. Study protocol.**
(DOCX)

**S3 File. Composition, chemistry, manufacturing, and controls of AYUSH 64.**
(DOCX)

**S4 File. Health and quality of life questionnaires (WHO-QOL & HR-BHF).**
(DOCX)

**S5 File. Additional data–standard of care drugs, site specific drugs & timelines, general linear model output, intention to treat analysis, and radiological data.**
(DOCX)

**S6 File. Additional data- adverse events.**
(DOCX)

**S7 File. Additional data- laboratory results.**
(DOCX)

**S8 File. Selected raw data—efficacy, withdrawals, adverse events.**
(DOCX)

## Acknowledgments

A special thanks to Vaidya Dr. Rajesh Kotecha, Secretary, Ministry of AYUSH, Government of India, for his invaluable guidance and encouragement towards the current study project and preparation of study publication. We are grateful to senior Vaidya KS Dhiman, former DG CCRAS, GOI for the speedy completion of the drug trial. We thank several research colleagues

in the CCRAS—Dr Ravindra Singh, Dr Shruti Khunduri, and Dr B S Sharma. We acknowledge with gratitude several colleagues from each trial site- Dr Manish Deshmukh, Dr Swati Munde, Dr Pratap Makhija, Dr Alia Rizvi, Prof Wahid Ali, Dr. Neeta Warty, Dr. Parth Dave and Mr Akash Saggam. We make a special mention of Dr Manesha Talekar, an Ayurvedic physician, who volunteered to be a co-investigator despite pregnancy and contracted COVID-19 during her medical duty. Dr Vinay Pawar played an important role in statistical analysis. Finally, we thank all the patients with folded hands for their participation and wholehearted support.

## Author Contributions

**Conceptualization:** Arvind Chopra, Girish Tillu, Manjit Saluja, Sanjeev Sarmukaddam, Narayanam Srikanth, Bhushan Patwardhan.

**Formal analysis:** Arvind Chopra, Girish Tillu, Sanjeev Sarmukaddam.

**Funding acquisition:** Narayanam Srikanth, Bhushan Patwardhan.

**Investigation:** Kuldeep Chuadhary, Govind Reddy, Alok Srivastava, Muffazal Lakdawala, Dilip Gode, Himanshu Reddy, Sanjay Tamboli, Manohar Gundeti, Ashwini Kumar Raut.

**Methodology:** Arvind Chopra, Girish Tillu, Manjit Saluja, Sanjeev Sarmukaddam, Ashwini Kumar Raut, Narayanam Srikanth, Bhushan Patwardhan.

**Project administration:** Arvind Chopra, Sanjay Tamboli, Manjit Saluja, B. C. S. Rao, Babita Yadav.

**Resources:** B. C. S. Rao, Babita Yadav, Narayanam Srikanth.

**Software:** Sanjay Tamboli.

**Supervision:** Arvind Chopra, Narayanam Srikanth, Bhushan Patwardhan.

**Validation:** Arvind Chopra, Kuldeep Chuadhary, Govind Reddy, Alok Srivastava, Muffazal Lakdawala, Dilip Gode, Himanshu Reddy, Sanjay Tamboli, Manjit Saluja, Manohar Gundeti, Ashwini Kumar Raut, B. C. S. Rao, Babita Yadav.

**Visualization:** Arvind Chopra.

**Writing – original draft:** Arvind Chopra, Girish Tillu.

**Writing – review & editing:** Girish Tillu, Kuldeep Chuadhary, Govind Reddy, Alok Srivastava, Muffazal Lakdawala, Dilip Gode, Himanshu Reddy, Sanjay Tamboli, Manjit Saluja, Sanjeev Sarmukaddam, Manohar Gundeti, Ashwini Kumar Raut, B. C. S. Rao, Babita Yadav, Narayanam Srikanth, Bhushan Patwardhan.

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
