## [Decision Letter · Decision Letter 0]

1 Jun 2022

PONE-D-21-24749

Coadministration of AYUSH 64 as an adjunct to Standard of Care in mild and moderate COVID-19: A randomised, controlled, multicentric clinical trial

PLOS ONE

Dear Dr. Chopra,

Thank you for submitting your manuscript to PLOS ONE. After careful consideration, we feel that it has merit but does not fully meet PLOS ONE’s publication criteria as it currently stands. Therefore, we invite you to submit a revised version of the manuscript that addresses the points raised during the review process. 

We look forward to receiving your revised manuscript.

Kind regards,

Katrien Janin, PhD

Staff Editor

PLOS ONE

Journal Requirements:

3. Thank you for stating the following in the Competing Interests section: "None of the authors have any financial conflict of interest regarding this study. The authors declare their relationship related to study as per the International Committee of Medical Journal Editors. Bhushan Patwardhan is Chairman, Interdisciplinary Ayush Research and Development Task Force on Covid-19. Kuldeep Chaudhary, Alok Srivastava, Govind Reddy, Manohar Gundeti, BCS Rao, Babita Yadav, Narayanam Srikanth work in Central Council for Research in Ayurvedic Sciences (CCRAS), Ministry of AYUSH (MoA), Government of India (GOI), New Delhi. Ashwinikumar Raut was a consultant for the study. Sanjay Tamoli was involved as a CRO. AYUSH-64 is a proprietary formulation of CCRAS."

**Comments to the Author**

1. Is the manuscript technically sound, and do the data support the conclusions?

Reviewer #1: Partly

Reviewer #2: Yes

2. Has the statistical analysis been performed appropriately and rigorously? 

Reviewer #1: I Don't Know

Reviewer #2: Yes

3. Have the authors made all data underlying the findings in their manuscript fully available?

Reviewer #1: No

Reviewer #2: Yes

4. Is the manuscript presented in an intelligible fashion and written in standard English?

Reviewer #1: Yes

Reviewer #2: No

5. Review Comments to the Author

Reviewer #1: The authors do not provide line numbers, so it is quite tedious to give specific edits. However, there are a good number of minor errors in English language usage throughout the manuscript. Given the standard to which the authors aspire, these should be corrected. There are verb tense issues in several places as well.

The authors do not provide raw data for evaluation. This seems indicated in what could be thought of as a somewhat contentious area.

The authors write "drugs such as hydroxychloroquine fell into disrepute". This is needless editorializing on a matter where there appears to be at least some good faith disagreement. There are certainly many western authors who would say the same about non-western systems of medicine, so it seems pointless to engage in such statements. It seems ironic that hydrochloroquine is then listed as part of standard of care later.

The authors write "It is now known that the exuberant and dysregulated immune response in COVID-19 leads to life threatening complications." Please provide citations and appropriately scope this statement with an estimate of the incidence of these in general populations.

The authors write "We used manually calculated raw scores (summation) for each domain". Why is "manually" stipulated here? Were computations performed by individuals and then entered into case report forms (CRFs)?

The authors write "A convenience study sample of 140 participants was finalized by AC and SS and considered

adequate to address the study research questions." This sounds quite ad hoc. What process was used to establish the convenience sample? What sampling biases were introduced by this process?

The statistical methods are described poorly, with a mix of jargon and software thrown together. Please describe the study endpoints and their analyses a little more clearly, perhaps grouping as seems fit to save tedious listing.

The actual structure of the analysis of variance (ANOVA) performed is unclear. Also, see below. The authors should provide a description that includes fixed effects, interaction terms, random effects (if any), and R-side correlation (if any). Linear model analysis must be accompanied by an assessment of model fit diagnostics, and corrective measures should be described in advance and described in the methods as necessary. All contrasts or estimates to be obtained should be described.

Generally, the analysis should be performed on both an intent-to-treat (ITT) and a per-protocol basis. The ITT analysis could be placed in a supplementary document.

Usual format for tables such as Table 1 are to footnote the p-values with the specific tests used.

Looking at Table 2 makes it clear that the data follow one of two structures:

* Fixed effect for treatment.

* Random effect for site.

* Potential random interaction of treatment and site.

* Repeated measures in cases where serial measurements are taken on a patient over time.

This structure can be handled easily within the linear mixed effects model framework. A consistent model should be used in all cases. If repeated measures are used, probably an unstructured correlation matrix will be most sensible and most flexible.

In Table 3 the authors use the footnote "*" to indicate "no difference"---the same footnote used before to indicate statistical significance. This is pointlessly confusing. Please use the standard approach that everyone else uses when reporting such results.

In Table 4 the authors note "(anchored with best and worst response)". What does this mean? This procedure is not described in the methods section.

Please modify Table 5 to industry norms.

The authors write "Despite serious concerns, the study arms were well matched on several measures including SOC." What are these concerns? Please demonstrate that these concerns are not an issue or please appropriately caveat the results of this trial.

The authors write "We believe that the current study has boldly addressed the need for evidence-based medicine to treat mild and moderate COVID-19." Doesn't this seem rather overblown?

The authors write "Several limitations were imposed by the chaotic and tragic pandemic situation. We encountered uncertainties and often contradictory advice regarding SOC and other COVID-19 related health

matters in the social and news media. During the first pandemic year, the patients were reluctant to seek medical care for fear of being stigmatized and this probably delayed the treatment for several patients as shown by the timelines in Table 1. Though, the primary efficacy was assessed by the attending physician in a blinded manner, a placebo response to some extent cannot be ruled out. Ayurveda is endearing to the Indian community. In view of absence of a-priori data, we settled for a convenience sample size for this study."

This further points up the need for a *very* clear description of how exactly this sample was selected. This is critical to the manuscript. Aside from that, this paragraph is overly dramatic and chatty. Everyone working in clinical trials had many, many issues during this time.

Please discuss the following potential limitations in much more detail:

* The study was open label and unblinded.

* The study sample was drawn as a "convenience sample".

Both of these issues have important implications for the interpretability of the results of the trial.

Reviewer #2: The reviewer’s comments on the article, ” Coadministration of AYUSH 64 as an adjunct to Standard of Care in mild and moderate COVID-19: A randomised, controlled, multicentric clinical trial” are as follows:

This is a multicentered clinical trial on AYUSH 64 as an adjunct to standard of care. The authors concluded that the AYUSH 64 group had shorter time to clinical recovery from randomization, higher proportion of recovery in the first week, and better performance in general health, quality of life, fatigue, anxiety, stress, sleep and other psychosocial metrics. In general, native-speaker English editing would be suggested since some sentences/words were difficult to understand. Besides, there were some points needed to explain in detail.

Abstract

In Results, please mention mean±95% CI days of 2 group.

Introduction

Reference 7 should be re-indexed. The authors may mention remdesivir, that is a commonly cited repurposing drug for COVID-19.

The authors may add some descriptions about the mechanisms that how AYUSH 64 works.

Methods

Is the sentence, “the duration of study was 12 weeks”, meant that for each participant, the treatment interval was 12 weeks? Please make the sentence clear.

Since AYUSH 64 is popularly used to treat acute onset febrile respiratory illnesses for over three decades in India, how did the authors prevent the participants to buy AYUSH 64 over the counter or on-line? In addition, how did the authors evaluate the compliance of both groups after discharge from hospitals?

Why and when did the study results be disseminated to the public through electronic social and print media, described in the section of subheading patient and public involvement?

In supplementary material, the protocol mentioned 3 regimen: i)AYUSH-64: 500 mg tablet, 2 tablets bid (twice daily) ii)Yashtimadhu: 300 mg tablet, 2 tablets bid (twice daily) iii)Sanshamani Vati Plus: Each tablet to contain 300 mg Guduchi plus 75 mg Pippali, 2 tablets bid (twice daily). Were the regimen ii, and iii used in this study?

Results

For CXR, moderate abnormalities in 2% patients in the AYUSH plus and 13% in the SOC were noted. Did the authors think patients in SOC had higher proportion of severe illness than those in AYUSH plus group?

On day 28, CXR showed mild abnormalities in 22% patients AYUSH plus and 21% patients SOC. Did the authors think that the residual abnormal CXR findings resolve later, after patients recovered clinically? Did the authors follow up the resolution of CXR findings?

In Table 2, * and ** deserved footnotes

In the subheading of adverse events in Table 5, the authors may change system organ classification to symptoms or diagnosis for clarification.

Discussions

The authors described “In our experience, despite sound medical advice, a large proportion of mild and moderate uncomplicated cases are admitted in the hospital and clog the system. AYUSH 64 plus SOC seemed to have significantly reduced the duration of hospitalization.” Please explain the government policy for admission and isolation of COVID-19 patients.

Since AYUSH 64 is popular in India and the study results was disseminated to the public through electronic social and print media, as per the authors’ description, could AYUSH 64 be so popular and could be bought over the counter? Did the authors consider the influence and deviation by SOC group who had taken AYUSH 64?

Did the authors consider that patients may take AYUSH 64 at home without admission in the future? For AYUSH 64 is popularly used to treat acute onset febrile respiratory illnesses for over three decades in India, why did the authors emphasize they need to be medically supervised?

6. PLOS authors have the option to publish the peer review history of their article (what does this mean?). If published, this will include your full peer review and any attached files.

Reviewer #1: No

Reviewer #2: **Yes: **Shu-Hsing Cheng

---

## [Author Response · Author response to Decision Letter 0]

19 Jul 2022

Reviewer #1: 

1)The authors do not provide line numbers, so it is quite tedious to give specific edits. However, there are a good number of minor errors in English language usage throughout the manuscript. Given the standard to which the authors aspire, these should be corrected. There are verb tense issues in several places as well.

Response: The line numbers are now provided in the revision. The entire text is reviewed by a native English language person and errors in grammar etc corrected.

2) The authors do not provide raw data for evaluation. This seems indicated in what could be thought of as a somewhat contentious area.

Response: This is now provided as a supporting document and further description for access to data also mentioned in the section on ‘data availability’.

3)The authors write "drugs such as hydroxychloroquine fell into disrepute". This is needless editorializing on a matter where there appears to be at least some good faith disagreement. There are certainly many western authors who would say the same about non-western systems of medicine, so it seems pointless to engage in such statements. It seems ironic that hydrochloroquine is then listed as part of standard of care later.

Response: The point is well taken and the sentence modified to indicate that the widespread use of HCQS in the earlier part of the pandemic in India was grossly restricted when several drug trials did not shown unequivocal evidence. Since submission of this manuscript in mid 2021, the Government of India has removed HCQS from the ‘standard of care’ and the same is now reflected in the revision. 

4) The authors write "It is now known that the exuberant and dysregulated immune response in COVID-19 leads to life threatening complications." Please provide citations and appropriately scope this statement with an estimate of the incidence of these in general populations.

Response: A citation is provided. There are no truly suitable prospective population studies to estimate the incidence of severe or life-threatening complications in the general population. However, we provide an estimate from hospital-based studies and a recent study published in PLOS ONE [: Grant MC, Geoghegan L, Arbyn M, Mohammed Z, McGuinness L, Clarke EL, et al. (2020) The prevalence of symptoms in 24,410 adults infected by the novel coronavirus (SARSCoV-2; COVID-19): A systematic review and meta-analysis of 148 studies from 9 countries. PLoS ONE 15(6): e0234765].

5) The authors write "We used manually calculated raw scores (summation) for each domain". Why is "manually" stipulated here? Were computations performed by individuals and then entered into case report forms (CRFs)?

Response: This was with reference to WHO Quality of Life Instrument- BREF. We followed the instructions contained in the WHO publication for calculating the domain and total score. Reference is provided in the main text. There were 27 questions belonging to 4 major domains. Each question was answered using a 5-point Likert response scale. Each of the 5 points of response (such as very poor, poor, neither poor nor good, good, and very good) were scored 1 to 5 - 1 was equivalent to very poor and 5 was equivalent to very good. The score for 3 questions reversed to follow the ascendancy order of the remaining questions so that higher scores meant better quality of life. A hard copy of the questionnaire was provided to each of the study subject for completion as per time points in the study protocol. The subject was asked to choose the likely response to each question and place a tick mark in the adjacent box showing the score of the response. The score of each question was added by the designated paramedic manually to calculate the score for each domain (manually calculated sum of raw scores in each domain); recorded in the questionnaire. The score for each domain and total for each participant was entered into the Excel based data sheet. 

6)The authors write "A convenience study sample of 140 participants was finalized by AC and SS and considered adequate to address the study research questions." This sounds quite ad hoc. What process was used to establish the convenience sample? What sampling biases were introduced by this process?

Response: We have rewritten the section. 

To find possible answers to address the study research questions and considering the logistics, a convenience study sample of 140 participants was felt to be adequate by AC (first author) and SS (Biostatistician co-author)”. This was also on the basis of available time & resources and relevant experience available then to carry out this trial. No formal estimation of required sample size for this study was attempted. 

7) The statistical methods are described poorly, with a mix of jargon and software thrown together. Please describe the study endpoints and their analyses a little more clearly, perhaps grouping as seems fit to save tedious listing.

Response: Please see the answer above in question No 6 above

8)The actual structure of the analysis of variance (ANOVA) performed is unclear. Also, see below. The authors should provide a description that includes fixed effects, interaction terms, random effects (if any), and R-side correlation (if any). Linear model analysis must be accompanied by an assessment of model fit diagnostics, and corrective measures should be described in advance and described in the methods as necessary. All contrasts or estimates to be obtained should be described.

Response: The statistical analysis and results are revised and shown in Table 2 and the primary efficacy measure result was revised as per the original plan. Being a 2 arm study, we adhere to the Student’s ‘t- test for independent samples and Mann Whitney statistic. However, to respond to the ‘random effect’ of sites, we have performed a general linear (mixed effect ) model for the primary efficacy measure and enclosed the results (SPSS output) in the supplement file (S5 File). 

9) Generally, the analysis should be performed on both an intent-to-treat (ITT) and a per-protocol basis. The ITT analysis could be placed in a supplementary document.

Response: The results of an ITT analysis on 137 participants who completed the randomization treatment phase are enclosed in the supplement file (S5 File)

10) Usual format for tables such as Table 1 are to footnote the p-values with the specific tests used.

Response: The format for Tables and foot notes is revised. 

11)Looking at Table 2 makes it clear that the data follow one of two structures:

* Fixed effect for treatment.

* Random effect for site.

* Potential random interaction of treatment and site.

* Repeated measures in cases where serial measurements are taken on a patient over time.

This structure can be handled easily within the linear mixed effects model framework. A consistent model should be used in all cases. If repeated measures are used, probably an unstructured correlation matrix will be most sensible and most flexible.

Response: Please see the answer to Question No 8. We have enclosed the results of a general linear (mixed effects) model.

12) In Table 3 the authors use the footnote "*" to indicate "no difference"---the same footnote used before to indicate statistical significance. This is pointlessly confusing. Please use the standard approach that everyone else uses when reporting such results.

Response: The Tables and foot notes have been revised to ensure uniform method of reporting data and in particular the ‘footnote’

13) In Table 4 the authors note "(anchored with best and worst response)". What does this mean? This procedure is not described in the methods section.

Response: The error is regretted. This pertains to a visual analogue scale and a a elaborate description is provided on the scoring technique of the two health and quality of life questionnaires used in the ‘methods section’. Table 4 and S4 File is modified accordingly.

14) Please modify Table 5 to industry norms.

Response: The suggestion is now incorporated and Table 5 is revised to show the AE are now classified as per symptom, sign and diagnosis along with the System organ classification .

15) The authors write "Despite serious concerns, the study arms were well matched on several measures including SOC." What are these concerns? Please demonstrate that these concerns are not an issue or please appropriately caveat the results of this trial.

Response: The word ‘serious’ was a wrong choice of word, and we acknowledge the error. It should be ‘important concerns. Accordingly, the text has been now been modified. There were several important a-priori concerns for the current drug trial study such as challenges of a drug trial during the upsurge of pandemic, fear and reluctance of COVID patients to participate, delay in the diagnosis and enrolment, different standards of care (COVID-19) at study sites, open label nature of AYUSH 64 intervention and placebo response, bias in the assessment of clinical recovery, and surreptitious use of AYUSH 64 and other Ayurvedic drugs. Several concerns were adequately addressed by the diligence of the study team, randomization procedure and treatment under direct observation (hospitalization phase). The study arms were well matched (Table 1). It seemed prudent to emphasize that hospitalization in the current study was responsible for good compliance and capture of robust data. The latter also made it possible to carry out a daily physical assessment of study participants by a competent COVID-19 physician to identify and endorse clinical recovery in a timely and systematic manner. Intense monitoring was also crucial to document the safety and tolerability of AYUSH 64. Importantly there were no clinically obvious drug interactions with the modern drugs. 16) The authors write "We believe that the current study has boldly addressed the need for evidence-based medicine to treat mild and moderate COVID-19." Doesn't this seem rather overblown?

Response: Yes. In hind sight the statement could be interpreted as an exaggeration. The statement is modified and reads ‘We believe that the current study has been able to provide some degree of evidence based medicine for the treatment mild and moderate COVID-19 (non emergency) using an integrative approach with an Ayurvedic drug (AYUSH 64) combined with standard modern medicine care’. 

17) The authors write "Several limitations were imposed by the chaotic and tragic pandemic situation. We encountered uncertainties and often contradictory advice regarding SOC and other COVID-19 related health matters in the social and news media. During the first pandemic year, the patients were reluctant to seek medical care for fear of being stigmatized and this probably delayed the treatment for several patients as shown by the timelines in Table 1. Though, the primary efficacy was assessed by the attending physician in a blinded manner, a placebo response to some extent cannot be ruled out. Ayurveda is endearing to the Indian community. In view of absence of a-priori data, we settled for a convenience sample size for this study." This further points up the need for a *very* clear description of how exactly this sample was selected. This is critical to the manuscript. Aside from that, this paragraph is overly dramatic and chatty. Everyone working in clinical trials had many, many issues during this time.

Response: We readily agree with the comment and incorporate the suggestion in the discussion . However, the issue at stake is complex and we have tried to express that in brief. Though it may be general knowledge , it may be appropriate to submit to the reader how challenging and complex drug trials were during the pandemic. The latter was more so during the initial six months or so during which time the current study was launched. It is prudent to add that some issues connected with drug trials were more regional than global. We believe that it is appropriate to state ‘ "Several limitations were imposed by the pandemic situation. We encountered uncertainties and often contradictory advice regarding SOC and other COVID-19 related health matters in the social and news media. During the first pandemic year, the patients were reluctant to seek medical care for fear of being stigmatized and this probably delayed the treatment for several patients as shown by the timelines in Table 1.’ We have further provided an informative reference from a recent PLOS ONE publication [: Chen Z, Chen L, Chen H (2021) The impact of COVID-19 on the clinical trial. PLoS ONE 16(5): e0251410]

With reference to the second half of the paragraph regarding physician centric issues and selection of participants, we acknowledge the need to elaborate the subject of selection of trial participants in the methods section. We first briefly introduce the then Government policy on treatment of COVID-19 which was followed. A new elaborate section on ‘Selection, Screening and Eligibility, and Management’ is now added under the methods section. 

A description of the ‘convenience sample size’ used in the current study is provided above under Question number 6 above. This is also clarified under the statistical design and analysis section of the manuscript.

18) Please discuss the following potential limitations in much more detail:

* The study was open label and unblinded.

* The study sample was drawn as a "convenience sample".

Both of these issues have important implications for the interpretability of the results of the trial.

Response: Agreed. Both the limitations were mentioned in the discussion section and are now modified for better clarity. The subject of sample size is also mentioned above in Question serial number 6 above. 

Reviewer #2: The reviewer’s comments on the article, ” Coadministration of AYUSH 64 as an adjunct to Standard of Care in mild and moderate COVID-19: A randomised, controlled, multicentric clinical trial” are as follows:

1) This is a multi-cantered clinical trial on AYUSH 64 as an adjunct to standard of care. The authors concluded that the AYUSH 64 group had shorter time to clinical recovery from randomization, higher proportion of recovery in the first week, and better performance in general health, quality of life, fatigue, anxiety, stress, sleep and other psychosocial metrics.

Response: This was the essence of the current study. It was a randomized controlled study. There were limitations too which were mostly related to convenience sample size and non-blinding nature intervention and several difficulties imposed by the COVID-19 pandemic on patient logistics and monitoring. All said and done, we believe that the outcome in favour of the Ayurvedic medicine AYUSH 64 to treat mild and moderate cases of COVID-19 along with standard care is encouraging. There is now community based observational data to support its standalone use also in such uncomplicated COVID-19 situations.

2) In general, native-speaker English editing would be suggested since some sentences/words were difficult to understand. Besides, there were some points needed to explain in detail.

Response: The point is well taken. We have corrected the manuscript with expert professional help for several grammatical and other errors and this is visible in the marked revision copy. Further, the sections on introduction, patient selection, trial procedures, and statistical design and analysis have been described in greater detail. Information on Remdesivir and other anti-COVID 19 drugs is also revised. Also, the discussion has been modified accordingly. 

3)Abstract

In Results, please mention mean±95% CI days of 2 group.

Response: We had incorporate the suggestion in the abstract.

4) Introduction

Reference 7 should be re-indexed. The authors may mention remdesivir, that is a commonly cited repurposing drug for COVID-19.

Response: yes. A reference has been added

5) The authors may add some descriptions about the mechanisms that how AYUSH 64 works.

Response: A brief description about the likely mechanism of action of AYUSH 64 is added in the discussion .

AYUSH 64 was not a classic Ayurveda formulation but contains medicinal plants described in classic texts and used by Ayurveda practitioners for several centuries in the Indian subcontinent. These medicinal plants were considered therapeutically useful to treat fever, pain and inflammation of varying aetiology and several medical disorders that include cough and breathing disorders and asthma. Several experimental studies, both in vitro and animal, have demonstrated some antiviral (herpes simplex, adenovirus and Chikungunya virus), anti-inflammatory and beneficial immunological effects of the medicinal plants in AYUSH 64. In a more recent in-silico molecular docking study, several ingredients of AYUSH 64 (and Akuammicine N-Oxide from Alstonia scholaris in particular) were found to bind well with the main protease enzyme of the SARS-CoV-2 and may well be an important mechanism of action. An uncontrolled exploratory clinical study of 38 patients suffering from influenza like illness and treated with AYUSH 64 for one week along with standard care showed no safety concerns and uncomplicated recovery and probably a lesser need of paracetamol and anti-allergic drugs. 

6) Methods

Is the sentence, “the duration of study was 12 weeks”, meant that for each participant, the treatment interval was 12 weeks? Please make the sentence clear.

Response: Thanks for this important observation. Each individual participant completed 12 weeks of study period and we have clarified the same in the text under methods.

7) Since AYUSH 64 is popularly used to treat acute onset febrile respiratory illnesses for over three decades in India, how did the authors prevent the participants to buy AYUSH 64 over the counter or on-line? 

Response: This is an important comment and we have provided an elaborate description under the ‘strengths and limitations’ section of the discussion. 

We were aware of a possibility of confounding by unauthorized use of drugs to treat COVID-19 by the study participants. All patients were counselled in detail prior to enrolment. A strict vigil and close monitoring during hospitalization phase was ensured by a study dedicated staff as per protocol. Following discharge, a close contact was kept with study participants using a special mobile phone app (see study methods) and telephonic reminders. However, it was difficult to vouch for total avoidance of other Ayurvedic drugs and or AYUSH 64 by study participants especially prior to enrolment or after discharge. 

It is prudent to add that as per the Government policy in vogue then, all study participants were admitted in Government run COVID dedicated hospitals for treatment irrespective of the severity of the illness. The latter ensured a strict direct observation of the study participants and compliance with all methods and procedures. Both the intervention groups were well matched for several variables at randomization baseline though our recorded did not show any prior Ayurvedic drug use. The study intervention was unblinded. It is unlikely that any drug other than that prescribed by the current study was used during in-patient treatment. Following discharge, patients were followed till week 12 completion and examined monthly. During the latter phase, patients from the AYUSH 64 plus standard care arm continued AYUSH 64 but not the patients from previous standard arm care. 

Ayurvedic drugs and Ashwagandha (Withania somnifera) and Guduchi (Tinospora cordifolia) in particular were used extensively both for prophylaxis and treatment of COVID-19 by the India population. Ayurvedic drugs were available over the counter and did not require a prescription. To begin with, AYUSH 64 was a lesser-known drug for use in COVID-19 in the earlier phase of the pandemic but in time was included in the Ayurveda drug promotional campaign by Government agencies. The current drug trial was begun on 18 June 20 and completed on 25 Oct 20. 

We have also described the measures taken by us to safeguard the study. 

Several measures were taken to safeguard against a surreptitious use of Ayurvedic drugs in the current study. Participants with a history of prior use of Ayurvedic drugs or any CAM were excluded. As mentioned above, the treatment during hospitalization was directly observed. A strict vigil was maintained by the nursing and paramedical staff who also recorded the consumption of AYUSH 64 as per protocol. Participants were closely monitored and examined during the follow up phase. A special mobile app was used for daily monitoring (see methods section above). Participants were repeatedly counselled against self-treatment. We believe that the surreptitious use of Ayurvedic medicines, if at all, was negligible in the current study and that the unauthorized use of AYUSH 64 was unlikely. On the other hand, the better health related outcome observed in the AYUSH Plus group during the pre-planned follow was indeed encouraging (Table 4).

8)In addition, how did the authors evaluate the compliance of both groups after discharge from hospitals?

Response: Patients randomized to AYUSH 64 plus standard care arm continued AYUSH 64 to complete 12 weeks of administration as per protocol. AYUSH 64 was provided free of cost to each patient and consumption monitored. Patients from the controlled arm of standard care continued non-specific treatment with drugs such as vitamins as per physician judgement and were not prescribed AYUSH 64. Patients were examined monthly and for long COVID symptoms and quality of life issues. In addition, patients were closely monitored using telephonic reminders and a specially designed mobile phone app (see methods). Some of the measures are described in the answer to Question No.7 above

9)Why and when did the study results be disseminated to the public through electronic social and print media, described in the section of subheading patient and public involvement?

Response: We have incorporated the excellent suggestion.A press release was issued by CCRAS, Ministry of AYUSH, Government of India, in May 2021 after completion of the current study. This was followed by a press meeting with selected leading news media to announce important results of the current study by the first author and other co-authors and experts . The management guidelines of COVID-19 issued by CCRAS from time to time were also updated with the key study results on the efficacy and safety of AYUSH 64. Simultaneously, all the principle investigators informed the outcome of the current study to each participant telephonically and those desirous were called for a meeting with the study physician. 

10) In supplementary material, the protocol mentioned 3 regimen: i)AYUSH-64: 500 mg tablet, 2 tablets bid (twice daily) ii)Yashtimadhu: 300 mg tablet, 2 tablets bid (twice daily) iii)Sanshamani Vati Plus: Each tablet to contain 300 mg Guduchi plus 75 mg Pippali, 2 tablets bid (twice daily). Were the regimen ii, and iii used in this study?

Response: We regret this confusion caused by the lack of clarity in the beginning of the ‘methods section’. We have now modified the introduction section to describe the initiative of AYUSH CCRAS Government of India under which three potential Ayurvedic drugs including AYUSH 64 were selected for independent randomized controlled drug trial studies using a different study team and study sites. However, there was a common protocol for methods which was registered in the Clinical Trial Registry of India and is enclosed as supplementary file (S2 File). All this is further clarified in the beginning of the ‘methods section’.

Here we only report the drug trial of AYUSH-64. 

To complete the answer, the other two drug trial studies were also completed in 2021 but the results are still under review with the sponsor.

11) Results- For CXR, moderate abnormalities in 2% patients in the AYUSH plus and 13% in the SOC were noted. Did the authors think patients in SOC had higher proportion of severe illness than those in AYUSH plus group? On day 28, CXR showed mild abnormalities in 22% patients AYUSH plus and 21% patients SOC. Did the authors think that the residual abnormal CXR findings resolve later, after patients recovered clinically? Did the authors follow up the resolution of CXR findings?

Response: We first regret the error in the data caused by lack of stringent review of the earlier radiographic data that was extracted from the case record forms and was not properly recorded in all cases. Also we found post submission, that no XRays were printed in one site and only the screening results were recorded. We thank you for the excellent comment and we now incorporate a more elaborate analysis and description.

Soon after submission, we completed a reassessment of the radiographic evaluation in the current study by a newly appointed independent radiologist. We have now revised the relevant sections pertaining to radiographic assessment.

At the outset, we concede that (i) Chest X-Rays were carried out only for diagnosis of atypical COVID-19 pneumonia and to ensure that there was no worsening on hospital discharge (ii) no standard protocol or scoring system was followed (iii) following discharge, X-Rays were only taken if the patient had a relapse of respiratory symptoms (iv) High resolution computed tomography scan was not taken as a routine or used to confirm resolution and absence of any complication such as pulmonary fibrosis (v) uncomplicated radiographic pneumonitis was labelled as moderate COVID-19 by the radiologist (vi) Digital X-Rays were taken at two sites (Mumbai and Nagpur) only and which are now used for detail analysis and inclusion in the current revised manuscript.

All digital X-Rays (90) were blinded for treatment allocation for the analysis report in the revision.

The revision contained the corrected results shown in S5 File (Supplement file).

In specific response to the Reviewer’s query, we agree that the radiographic severity (definite pneumonitis) of COVID-19 seems to be somewhat more in the standard care group though it may not be of clinical significance. Also, there were other caveat that we explain. We also agree with the reviewer’s observation that radiographic resolutions may take much longer than clinical recovery and we have included a citation from a research study. In hindsight, we should have followed participants with abnormal X-Rays during hospitalization with repeated radiological evaluation to document radiographic resolution, but we were discouraged by small numbers of participants with abnormal X-Ray on discharge and logistics and study budget. Both the groups were well balanced and patients were closely monitored during randomization treatment and during follow up. Only in one case, there was rapid clinical and radiologic deterioration of respiratory status warranting ICU care and oxygen administration and withdrawal from the study-the patient made complete recovery and belonged to standard care intervention arm in the study..

As regards a more serious nature of illness in the SOC group we submit the following statement in the discussion section. But also add that in view of the limitations, the reader needs to exercise caution while interpreting the radiographic results

More participants in the SOC arm showed definite radiographic pneumonitis (30% versus 19%) and also a higher proportion of failure of resolution (85% versus 63%) at the time of clinical recovery/hospital discharge; one patient with mild radiographic disease in SOC rapidly developed rapidly progressive respiratory distress and required oxygenation and intensive care for recovery. All this may suggest a more serious disease in the SOC arm but this does not seem to have been the case as shown by several other clinical variables, serum biomarkers and overall standard of care drugs used and the treatment response in the current study (Table 1. Table 3, and S5 File). However, it is important to concede that neither a uniform standard protocol nor CT scan imaging was used to evaluate chest skiagrams and/or resolution and thus any interpretation of the current findings of resolution may be premature and requires utmost caution. Radiographic abnormalities may persist beyond clinical recovery and take a longer time for complete resolution [38]. Also, there is insufficient data on prospective evaluation of conventional radiography in COVID-19 [38]. 

12) In Table 2, * and ** deserved footnotes

Response: Table 2 has been revised, also in view of comments of Reviewer 1.

13) In the subheading of adverse events in Table 5, the authors may change system organ classification to symptoms or diagnosis for clarification.

Response: Earlier we used the WHO organ classification system. The AE are not mentioned as symptoms or diagnosis

14)Discussions- The authors described “In our experience, despite sound medical advice, a large proportion of mild and moderate uncomplicated cases are admitted in the hospital and clog the system. AYUSH 64 plus SOC seemed to have significantly reduced the duration of hospitalization.” Please explain the government policy for admission and isolation of COVID-19 patients.

Response: We have revised our statement. Firstly, all COVID-19 cases were admitted in the hospital during the early pandemic as per the recommendations of the Government. This policy was quickly revised as the burden was overwhelming and mild and moderate cases were to monitored in a domiciliary setting or quarantine facility.. However, in the current study, all patients were admitted as per Government recommendations for COVID-19 but seemed to have improved the quality of clinical monitoring and capture of robust data. The results of the current study show that addition of AYUSH hastened clinical recovery and reduced the hospital period. Mild and moderate COVID-19 does not require hospitalization unless there are risk factors and/or likelihood of disease progression or complications. In the study we only treated mild and moderate cases of COVID-19 and the observations of the current study cannot be extrapolated to severe and complicated cases. 

15) Discussion-Since AYUSH 64 is popular in India and the study results was disseminated to the public through electronic social and print media, as per the authors’ description, could AYUSH 64 be so popular and could be bought over the counter? Did the authors consider the influence and deviation by SOC group who had taken AYUSH 64?

Did the authors consider that patients may take AYUSH 64 at home without admission in the future? For AYUSH 64 is popularly used to treat acute onset febrile respiratory illnesses for over three decades in India, why did the authors emphasize they need to be medically supervised?

Response: The above comment is well taken and accordingly the current revision contains appropriate clarifications and references on the subject. Please refer to the response described in Question serial 6 , 7 and 9 which also addresses a large part of this question. Undoubtedly, Ayurvedic drugs including AYUSH 64 were available across-the-counter without any restrictions. None of the participant patients stated having taken Ayurvedic drug prior to hospital admission but this cannot be vouched for veracity. Patients may have taken drugs from their general practitioner without knowledge of the name or nature (drug) as is often the practise in India. Any effect of AYUSH 64 or Ayurvedic drugs taken prior to admission hopefully will be balanced in the two study treatment arms by randomization. Post discharge, the arm with continued AYUSH 64 did better than the arm without AYUSH 64 in several measures of quality of life and did not differ for long COVID-19 or any other COVID related complication. It is difficult to exclude with certainty the effect of a surreptitious use of Ayurvedic drug or AYUSH 64 by some participants in the post-discharge follow up phase but this would not be of relevance to at least the primary efficacy measure. Compliance during the study is discussed in response to Question number 8.

---

## [Decision Letter · Decision Letter 1]

1 Sep 2022

PONE-D-21-24749R1Coadministration of AYUSH 64 as an adjunct to Standard of Care in mild and moderate COVID-19: A randomized, controlled, multicentric clinical trialPLOS ONE

Dear Dr. Chopra

Thank you for submitting your manuscript to PLOS ONE. After careful consideration, we feel that it has merit but does not fully meet PLOS ONE’s publication criteria as it currently stands. Therefore, we invite you to submit a revised version of the manuscript that addresses the points raised during the review process.

We look forward to receiving your revised manuscript.

Kind regards,

Shu-Hsing Cheng, Ph.D.

Guest Editor

PLOS ONE

Journal Requirements:

Additional Editor Comments:

I will suggest major revision according to 2 reviewers’ comments. Especially, strength and limitation should focus on the methods and data interpretation.

Reviewers' comments:

Reviewer's Responses to Questions

**Comments to the Author**

1. If the authors have adequately addressed your comments raised in a previous round of review and you feel that this manuscript is now acceptable for publication, you may indicate that here to bypass the “Comments to the Author” section, enter your conflict of interest statement in the “Confidential to Editor” section, and submit your "Accept" recommendation.

Reviewer #1: (No Response)

Reviewer #3: All comments have been addressed

2. Is the manuscript technically sound, and do the data support the conclusions?

Reviewer #1: (No Response)

Reviewer #3: Yes

3. Has the statistical analysis been performed appropriately and rigorously? 

Reviewer #1: (No Response)

Reviewer #3: Yes

4. Have the authors made all data underlying the findings in their manuscript fully available?

Reviewer #1: (No Response)

Reviewer #3: Yes

5. Is the manuscript presented in an intelligible fashion and written in standard English?

Reviewer #1: (No Response)

Reviewer #3: No

6. Review Comments to the Author

Reviewer #1: In general, most issues have been dealt with in the response. The manuscript is improved. The following are less important issues:

There still are many minor issues with English language usage throughout the manuscript. These are not critical, but do detract from the presentation.

Lines 163-164 seem quite odd. Perhaps these are not correctly expressing the authors' desired meaning?

The statistical methods section is somewhat improved with regard to content. However, the English language usage requires a lot of rewriting.

It appears that the authors ran both a fixed effects model (with treatment and site as fixed effects) and a linear mixed effects model (with treatment as a fixed effect and site as a random effect). Is this correct? If so, why? The mixed effects model is arguably better if the authors wish to generalize to "all sites" --- assuming these sites can be thought of as a representative sample if "all sites". The fixed effects model is appropriate if the authors wish to restrict their inference to "only the sites studied".

For tabulation, please footnote "not significant" results with "NS" and not with "*". Use the following scheme:

* Statistically significant (p < 0.05)

NS Not statistically significant (p >= 0.05)

a Other footnote that does not relate to statistical significance

b Other footnote that does not relate to statistical significance

etc.

Table 2 requires some reformatting for readability, in particular the rows with the p-values. I was not able to 100% reproduce the values in the table with the raw data --- perhaps there are some data exclusions that are relevant? However, the results are fairly close.

The last column of Table 2 is incorrect. This should show the result of a model that incorporates site as a random variable. A simple t-test is incorrect for the entire data set.

Please see the following code which reproduces part of Table 2 as well as demonstrating how to perform the linear mixed effects analysis using the R programming language.

# Load packages.

library(tidyverse)

library(broom)

library(lme4)

library(car)

# Read data and set data conventions.

S8 <- read_csv("S8_Raw.csv")

names(S8) <- c("PatientID", "Group", "Age", "Gender", "Site", "RtoCR", "SOtoCR")

S8 <- S8 %>%

mutate(Site = str_c("Site ", Site))

#-------------------------------------------------------------------------------

# Perform brief evaluation of time from randomization to complete recovery.

# Plot the data.

S8 %>%

ggplot(aes(x = Group, y = RtoCR)) +

geom_boxplot() +

facet_grid( ~ Site)

S8 %>%

ggplot(aes(x = Group, y = RtoCR)) +

geom_boxplot() +

facet_grid( ~ Site) +

scale_y_log10()

S8 %>% ggplot(aes(x = Age, y = RtoCR, col = Group)) +

geom_point() +

facet_grid(~ Site)

S8 %>% ggplot(aes(x = Gender, y = RtoCR, col = Group)) +

geom_boxplot() +

facet_grid(~ Site)

S8 %>% ggplot(aes(x = SOtoCR, y = RtoCR, col = Group)) +

geom_point() +

facet_grid(~ Site)

# Perform t-tests within each site.

Stats <- S8 %>%

group_by(Site) %>%

nest() %>%

mutate(

fit = map(data, ~t.test(.x$RtoCR ~ .x$Group)),

tidy = map(fit, tidy)

) %>%

unnest(tidy)

Stats

# Perform linear mixed effect analysis with site as a random effect.

fit <- lmer(RtoCR ~ Group + (1 | Site), data = S8)

#fit <- lmer(log(RtoCR) ~ Group + (1 | Site), data = S8)

plot(fit)

Anova(fit, test = "F")

# NOTE Arguably slightly better fit on logarithmic scale with no change in

# conclusion. It is easiest to interpret on the original scale.

#-------------------------------------------------------------------------------

# Perform brief evaluation of time from symptom onset to complete recovery.

# Plot the data.

S8 %>%

ggplot(aes(x = Group, y = SOtoCR)) +

geom_boxplot() +

facet_grid( ~ Site)

S8 %>%

ggplot(aes(x = Group, y = SOtoCR)) +

geom_boxplot() +

facet_grid( ~ Site) +

scale_y_log10()

S8 %>% ggplot(aes(x = Age, y = SOtoCR, col = Group)) +

geom_point() +

facet_grid(~ Site)

S8 %>% ggplot(aes(x = Gender, y = SOtoCR, col = Group)) +

geom_boxplot() +

facet_grid(~ Site)

# Perform t-tests within each site.

Stats <- S8 %>%

group_by(Site) %>%

nest() %>%

mutate(

fit = map(data, ~t.test(.x$SOtoCR ~ .x$Group)),

tidy = map(fit, tidy)

) %>%

unnest(tidy)

Stats

# Perform linear mixed effect analysis with site as a random effect.

fit <- lmer(SOtoCR ~ Group + (1 | Site), data = S8)

fit <- lmer(log(SOtoCR) ~ Group + (1 | Site), data = S8)

plot(fit)

Anova(fit, test = "F")

# NOTE Better fit on logarithmic scale but again with no change in overall

# conclusion. It is easiest to interpret on the original scale.

Reviewer #3: This manuscript described coadministration of AYUSH 64 as an adjunct to Standard of Care in mild and moderate COVID-19 in India: A randomized, controlled, multicentric clinical trial.

The study explored the efficacy of AYUSH 64, a standard polyherbal Ayurvedic drug in mild and moderate COVID-19. The result presents AYUSH 64 in combination with SOC hastened recovery, reduced hospitalization, and improved health in COVID-19. It was considerably safe and well-tolerated.

In general, this is a fair-written manuscript. Other points in this manuscript needed to be clarified are listed below:

Major revisions:

1. In Method, line 145 and 220: open label study is one of selection bias in your study, and your primary endpoint and secondary points are subjective, not scientific or medical specific term. How did you measure RT-PCR of SARS CoV-2? Roche Cobas? What is the definition of negative? Did every center use the same machine? or central lab was provided? If not, you might have information bias in analyzing outcome.

2. Standard of care, line 202: there are several listed medications in supplement file, and dexamethasone was proved effective treatment in COVID-19, except remdesvir. Could you provide the proportion of dexamethasone used in two arms? I think the data would impact the outcome.

3. In Discussion, strength and limitation, line 566: A convenience study sample of 140 participants was felt to be adequate by AC (first author) and SS (Biostatistician co-author)”, and no formal estimation of required sample size for this study was attempted. I think that the authors might write in paragraph of limitations of conclusion, because sampling bias or selection bias is possible. Moreover, the authors sould list some bias in terms of limitations, not to describe the chaos situation when the study was conducted during the pandemic.

Minor revisions:

1. Discussion, line 470: completer analysis should be corrected as per protocol analysis.

2. Reference 50 and 52, line 925 and 935: volume and page are missed.

7. PLOS authors have the option to publish the peer review history of their article (what does this mean?). If published, this will include your full peer review and any attached files.

Reviewer #1: No

Reviewer #3: **Yes: **CHIEN-YU CHENG

---

## [Author Response · Author response to Decision Letter 1]

4 Oct 2022

PONE-D-21-24749R1

Coadministration of AYUSH 64 as an adjunct to Standard of Care in mild and moderate COVID-19: A randomized, controlled, multicentric clinical trial

PLOS ONE

Response to Author/Referee Comments on the First Revision dated 02 Sept 2022. 

Please Note: The author response is shown below each comment as provided by the journal editor.

Comments to the Author and Response

I) General:

1. If the authors have adequately addressed your comments raised in a previous round of review and you feel that this manuscript is now acceptable for publication, you may indicate that here to bypass the “Comments to the Author” section, enter your conflict of interest statement in the “Confidential to Editor” section, and submit your "Accept" recommendation.

Reviewer #1: (No Response)

Reviewer #3: All comments have been addressed

Author: No comment

2. Is the manuscript technically sound, and do the data support the conclusions?

Reviewer #1: (No Response)

Reviewer #3: Yes

Author: No comment

3. Has the statistical analysis been performed appropriately and rigorously? 

Reviewer #1: (No Response)

Reviewer #3: Yes

Author: No comment

4. Have the authors made all data underlying the findings in their manuscript fully available?

Reviewer #1: (No Response)

Reviewer #3: Yes

Author : Further data has been added to address the comment on the efficacy analysis (below) 

5. Is the manuscript presented in an intelligible fashion and written in standard English?

Reviewer #1: (No Response)

Reviewer #3: No

Author: The revision was suitably edited by professional in English language. However, we note (see below) that there are still ‘minor issues with English language usage throughout the manuscript’ according to Reviewer 1. The current revision has been edited by a new professional in English language.

6. Review Comments to the Author

Author: we are grateful to the author and the referees for their comments to improve the manuscript

II) Reviewer #1: In general, most issues have been dealt with in the response. The manuscript is improved. 

The following are less important issues:

1)There still are many minor issues with English language usage throughout the manuscript. These are not critical but do detract from the presentation.

Author: This revision has been checked by a new professional in English language use. 

2)Lines 163-164 seem quite odd. Perhaps these are not correctly expressing the authors' desired meaning?

Author: Deleted

3)The statistical methods section is somewhat improved with regard to content. However, the English language usage requires a lot of rewriting.

Author: This has been again checked and revised by the study Biostatistician. We followed the a-priori statistical plan as per protocol. However, some additional analysis was performed in response to the excellent suggestion of the referee.

4) It appears that the authors ran both a fixed effects model (with treatment and site as fixed effects) and a linear mixed effects model (with treatment as a fixed effect and site as a random effect). Is this correct? If so, why? The mixed effects model is arguably better if the authors wish to generalize to "all sites" --- assuming these sites can be thought of as a representative sample if "all sites". The fixed effects model is appropriate if the authors wish to restrict their inference to "only the sites studied".

Author: (a) The data on the primary efficacy measure ‘time from randomization to clinical recovery’ in the study between the two study groups was analysed using Students’ t test and MW statistic and the results are shown in Table 2.

(b) We further performed a linear mixed effect model (treatment as fixed effect and site as random) as recommended by the referee the results were consistent with our earlier primary efficacy analysis. This is mentioned in the text and the detail model output is shown in Supplementary 5 File (Table S5.4). 

5)For tabulation, please footnote "not significant" results with "NS" and not with "*". Use the following scheme:

* Statistically significant (p < 0.05)

NS Not statistically significant (p >= 0.05)

a Other footnote that does not relate to statistical significance

b Other footnote that does not relate to statistical significance

etc.

Author: We have used this scheme for the Tables and we thank the referee

6) Table 2 requires some reformatting for readability, in particular the rows with the p-values.

Author: This has been done.

7)I was not able to 100% reproduce the values in the table with the raw data --- perhaps there are some data exclusions that are relevant? However, the results are fairly close.

Author: Please note that the raw data of 136 subjects in the earlier supplement pertained to per protocol analysis. Now we have provided the data of 140 subjects out of which 137 qualify for ITT analysis (3 withdrawn prematurely and did not provide data on clinical recovery).

We have reanalysed both per protocol and ITT analysis using the raw data provided in the supplement. Overall there are no variations from the earlier results.. 

8) The last column of Table 2 is incorrect. This should show the result of a model that incorporates site as a random variable. A simple t-test is incorrect for the entire data set.

Author: As per protocol, we first carried out a standard head-to-head comparison of the two study intervention groups using Students T test and MW statistic and wish to retain this as the primary efficacy measure analysis . However, we carried out a liner mixed effect model to exclude the effect of study site on primary outcome as suggested by the referee and include the result in the text. 

9)Please see the following code which reproduces part of Table 2 as well as demonstrating how to perform the linear mixed effects analysis using the R programming language.

# Load packages.

library(tidyverse)

library(broom)

library(lme4)

library(car)

# Read data and set data conventions.

S8 <- read_csv("S8_Raw.csv")

names(S8) <- c("PatientID", "Group", "Age", "Gender", "Site", "RtoCR", "SOtoCR")

S8 <- S8 %>%

mutate(Site = str_c("Site ", Site))

#-------------------------------------------------------------------------------

# Perform brief evaluation of time from randomization to complete recovery.

# Plot the data.

S8 %>%

ggplot(aes(x = Group, y = RtoCR)) +

geom_boxplot() +

facet_grid( ~ Site)

S8 %>%

ggplot(aes(x = Group, y = RtoCR)) +

geom_boxplot() +

facet_grid( ~ Site) +

scale_y_log10()

S8 %>% ggplot(aes(x = Age, y = RtoCR, col = Group)) +

geom_point() +

facet_grid(~ Site)

S8 %>% ggplot(aes(x = Gender, y = RtoCR, col = Group)) +

geom_boxplot() +

facet_grid(~ Site)

S8 %>% ggplot(aes(x = SOtoCR, y = RtoCR, col = Group)) +

geom_point() +

facet_grid(~ Site)

# Perform t-tests within each site.

Stats <- S8 %>%

group_by(Site) %>%

nest() %>%

mutate(

fit = map(data, ~t.test(.x$RtoCR ~ .x$Group)),

tidy = map(fit, tidy)

) %>%

unnest(tidy)

Stats

# Perform linear mixed effect analysis with site as a random effect.

fit <- lmer(RtoCR ~ Group + (1 | Site), data = S8)

#fit <- lmer(log(RtoCR) ~ Group + (1 | Site), data = S8)

plot(fit)

Anova(fit, test = "F")

# NOTE Arguably slightly better fit on logarithmic scale with no change in

# conclusion. It is easiest to interpret on the original scale.

#-------------------------------------------------------------------------------

# Perform brief evaluation of time from symptom onset to complete recovery.

# Plot the data.

S8 %>%

ggplot(aes(x = Group, y = SOtoCR)) +

geom_boxplot() +

facet_grid( ~ Site)

S8 %>%

ggplot(aes(x = Group, y = SOtoCR)) +

geom_boxplot() +

facet_grid( ~ Site) +

scale_y_log10()

S8 %>% ggplot(aes(x = Age, y = SOtoCR, col = Group)) +

geom_point() +

facet_grid(~ Site)

S8 %>% ggplot(aes(x = Gender, y = SOtoCR, col = Group)) +

geom_boxplot() +

facet_grid(~ Site)

# Perform t-tests within each site.

Stats <- S8 %>%

group_by(Site) %>%

nest() %>%

mutate(

fit = map(data, ~t.test(.x$SOtoCR ~ .x$Group)),

tidy = map(fit, tidy)

) %>%

unnest(tidy)

Stats

# Perform linear mixed effect analysis with site as a random effect.

fit <- lmer(SOtoCR ~ Group + (1 | Site), data = S8)

fit <- lmer(log(SOtoCR) ~ Group + (1 | Site), data = S8)

plot(fit)

Anova(fit, test = "F")

# NOTE Better fit on logarithmic scale but again with no change in overall

# conclusion. It is easiest to interpret on the original scale.

Author: We thank the Referee for his very considerate suggestion and providing the code for R programming. We have no experience with R. We have used standard statistical programs mentioned in the text and hope that the same will be accepted.

III) Reviewer #3: This manuscript described coadministration of AYUSH 64 as an adjunct to Standard of Care in mild and moderate COVID-19 in India: A randomized, controlled, multicentric clinical trial.

The study explored the efficacy of AYUSH 64, a standard polyherbal Ayurvedic drug in mild and moderate COVID-19. The result presents AYUSH 64 in combination with SOC hastened recovery, reduced hospitalization, and improved health in COVID-19. It was considerably safe and well-tolerated. In general, this is a fair-written manuscript. Other points in this manuscript needed to be clarified are listed below:

Author: We thank the referee.

Major revisions:

1). (a) In Method, line 145 and 220: open label study is one of selection bias in your study, and (b) your primary endpoint and secondary points are subjective, not scientific or medical specific term.(c) How did you measure RT-PCR of SARS CoV-2? Roche Cobas? What is the definition of negative? Did every center use the same machine? or central lab was provided? If not, you might have information bias in analyzing outcome. 

Author: 

(a) Agreed and we have included the possibility of a selection bias due to the open label nature of the study and convenience sample in the study limitation. We have further explained the good match for several variables at randomization baseline. Also, the inpatients were directly observed for drug administration, monitoring of symptoms and oximetry, clinical recovery and adverse events by a dedicated study team. The primary efficacy outcome was assessed blindly by a dedicated independent COVID physician.

 (b) We agree that the primary and secondary end points are dependent upon the clinical response elicited from the patient and prone to inaccuracy and bias. But in our case, there was no single criteria but a set of criteria to determine ‘clinical recovery’ (CR) within a time frame of 28 days. CR was a combination of resolution of symptoms and normal peripheral oximetry for 48 hours and a negative RT-PCR assay. We wish to add that the concept of CR in clinical drug trials for COVID-19 was still evolving when we designed this study in May-June 2020. Interestingly, there continues to be paucity of data on the clinical and microbial efficacy endpoints and what constitutes CR in controlled drug trial studies of mild to moderate COVID-19 in published literature. Most of the studies have used ‘viral load’ as the primary efficacy as a surrogate marker of clinical improvement. The WHO index of severity of 7 symptoms to indicate improvement at some fixed time points between 7-14 days after intervention were often used as secondary efficacy measure. There is no universally accepted definition of clinical recovery in mild to moderate COVID-19. Some studies have used clinical end points of hospitalization, morbidity or mortality. In this regard, our primary efficacy measure was unique. It was more wholesome, pragmatic and reasonable and based on our early clinical experience with mild moderate cases of COVID 19. It should be noted that we applied the criteria in hospitalized patients where the clinical monitoring was daily, direct and in real time. There may be more subjectivity in a domiciliary setting or when clinical data was obtained telephonically or through digital platform. It is prudent to add, that only one patient in the study progressed to severe stage. Also, during the follow up of 12 weeks after clinical recovery, there were no clinical concerns of any important COVID sequel. However, encouraged by the referee comment, we have included a brief note on this pivotal issue in discussion section. 

(c) None of the labs at study sites used Roche Cobas and we did not use a central lab. 

This was a Government of India sponsored study carried out at Government approved medical facilities. It was mandatory for the labs to follow the instructions issues by Indian Council of Medical Research (ICMR) for the specific SARS-CoV-2 RT-PCR assay 

[Ref: (i) https://www.icmr.gov.in/ctestlab.html

(ii) Advisory_for_Reagents_TestingLabs_v1.pdf (icmr.gov.in) 

(iii) Microsoft Word - SARS_CoV2_using_TaqPath_COVID19_ComboKit (icmr.gov.in) ]. 

All kits, reagents and equipment was to conform to the recommendations of ICMR. Since then, we have confirmed that the labs at each of the three study sites followed ICMR guidelines of quality control and standard testing but procured reagents and equipment from different manufacturers. Each of the site lab followed the manufacturer guidelines for reporting the results of the RT-PCR assay. A negative RT-PCR was reported usually after a run of 28-30 cycles. 

However, we admit that some information bias is likely in reporting outcome based on RT-PCR assay in the current study and this is now included in the manuscript.

2). Standard of care, line 202: there are several listed medications in supplement file, and dexamethasone was proved effective treatment in COVID-19, except remdesvir. Could you provide the proportion of dexamethasone used in two arms? I think the data would impact the outcome.

Author: We have listed the medications in the supplement file that were accepted as standard of care according to the guidelines for treatment issued by Indian Council of Medical Research and Ministry of Health, Government of India during the pandemic. This is referenced in the paper. However, the drugs were to be used as per severity of the illness and clinical discretion. There were differences in the use of drugs at study sites. Dexamethasone was only used in one subject who developed an acute onset of breathlessness and drop in arterial oxygen; withdrawn from the study. 

3. In Discussion, strength and limitation, line 566: A convenience study sample of 140 participants was felt to be adequate by AC (first author) and SS (Biostatistician co-author)”, and no formal estimation of required sample size for this study was attempted. I think that the authors might write in paragraph of limitations of conclusion, because sampling bias or selection bias is possible. Moreover, the authors should list some bias in terms of limitations, not to describe the chaos situation when the study was conducted during the pandemic.

Author: We agree with the comment and accordingly modified the section on limitations and conclusion. We have mentioned the possibility of a selection bias as described in the comment above also/

Minor revisions:

1). Discussion, line 470: completer analysis should be corrected as per protocol analysis.

Author: We have corrected the term

2. Reference 50 and 52, line 925 and 935: volume and page are missed.

Author: The references have been corrected as per the citation provided by the published paper and conforming to PLOS One.

---

## [Editor Report · Decision Letter 2]

10 Oct 2022

PONE-D-21-24749R2Coadministration of AYUSH 64 as an adjunct to Standard of Care in mild and moderate COVID-19: A randomized, controlled, multicentric clinical trialPLOS ONE

Dear Dr. Chopra,

Thank you for submitting your manuscript to PLOS ONE. After careful consideration, we feel that it has merit but does not fully meet PLOS ONE’s publication criteria as it currently stands. Therefore, we invite you to submit a revised version of the manuscript that addresses the points raised during the review process.

We look forward to receiving your revised manuscript.

Kind regards,

Shu-Hsing Cheng, Ph.D.

Guest Editor

PLOS ONE

Journal Requirements:

Additional Editor Comments (if provided):

The whole manuscript still contained many grammar errors and missing notes. For examples:

1. At line 39, “Significance p <0.05, two sided.” This is not a complete sentence

2. In the abstract, “App.”, “CI”, “PP”, abbreviations were presented without preceding full names.

3. At line 61, “ the World” should the “the world”

4. At line 97, HCQS without full names

5. At line 163, the meanings of “The patients did not assess the burden or the outcome of the intervention.” was not clear.

6. At line 171, “ real life PCR” must be “real time”

7. At line 293, CRO were presented without preceding full name.

8. For figures and tables:

Table 1 Footnote: (p,0.05) may be p<0.05, like other footnotes in table 2-5.

Table 2 # was not explained in the footnote.

Table 3: * did not explain the comparison groups or arms. Is it significant?

Table 4: ** was not explained in the footnote.

Table 5: In Column 1, is “Investigation” represented anything?

Figure 2: “completer” was suggested change to “per protocol”

9. At line 417-418, “there was a significant reduction in serum biomarkers of COVID-19 in each of the study groups without any significant difference (Table 3).” This sentence was not clear.

10. At line 430-432, “ In comparison to SOC, AYUSH Plus showed significant improvement in several domains (physical health, psychological health, social relationship, and environmental well-being) in the WHO QOL BREF and the total HR-BHF score the time of clinical recovery and pre-determined follow-up time points (Table 4).” May the authors check again the completeness of the sentences?

Most of the contents had been revised according to previous 3 reviewers’ comments. However, the whole manuscript needs dedicated edition to improve the readability and quality.
---

## [Author Response · Author response to Decision Letter 2]

21 Oct 2022

Additional Editor Comments (if provided):

The whole manuscript still contained many grammar errors and missing notes. For examples:

Response: My apologies for these errors. The manuscript has been rechecked by a profession in use of English language and medical writing.

Editor:

1. At line 39, “Significance p <0.05, two sided.” This is not a complete sentence

Response: Corrected

2. In the abstract, “App.”, “CI”, “PP”, abbreviations were presented without preceding full names.

Response: All abbreviations are rechecked and full name provided

3. At line 61, “ the World” should the “the world”

Response: corrected

4. At line 97, HCQS without full names

Response: corrected

5. At line 163, the meanings of “The patients did not assess the burden or the outcome of the intervention.” was not clear

Response: This is deleted and the sections rewritten.

6. At line 171, “ real life PCR” must be “real time”

Response: corrected

7. At line 293, CRO were presented without preceding full name.

Response: A full form was provided in the earlier text

8. For figures and tables:

Table 1 Footnote: (p,0.05) may be p<0.05, like other footnotes in table 2-5.

Table 2 # was not explained in the footnote.

Table 3: * did not explain the comparison groups or arms. Is it significant?

Table 4: ** was not explained in the footnote.

Table 5: In Column 1, is “Investigation” represented anything?

Figure 2: “completer” was suggested change to “per protocol”

Response: All Tables are rechecked and corrections made. Figure 2 is revised

9. At line 417-418, “there was a significant reduction in serum biomarkers of COVID-19 in each of the study groups without any significant difference (Table 3).” This sentence was not clear.

Response: Though the difference from baseline was significant in each of the two groups, there was no significant difference between the groups. This is now further clarified 

10. At line 430-432, “ In comparison to SOC, AYUSH Plus showed significant improvement in several domains (physical health, psychological health, social relationship, and environmental well-being) in the WHO QOL BREF and the total HR-BHF score the time of clinical recovery and pre-determined follow-up time points (Table 4).” May the authors check again the completeness of the sentences?

Response: The sentence is further made clear. It now reads ‘In comparison to SOC, AYUSH Plus showed significant improvement in several domains (physical health, psychological health, social relationship, and environmental well-being) in the WHO QOL BREF and the total HR-BHF score at the time of clinical recovery and during follow-up (Table 4).’

---

## [Editor Report · Decision Letter 3]

22 Feb 2023

Coadministration of AYUSH 64 as an adjunct to Standard of Care in mild and moderate COVID-19: A randomized, controlled, multicentric clinical trial

PONE-D-21-24749R3

Dear Dr. Arvind Chopra

We’re pleased to inform you that your manuscript has been judged scientifically suitable for publication and will be formally accepted for publication once it meets all outstanding technical requirements.

Kind regards,

Shu-Hsing Cheng, Ph.D.

Guest Editor

PLOS ONE

Additional Editor Comments (optional):

This version is comprehensive, and previleged to be accepted.
---

## [Editor Report · Acceptance letter]

6 Mar 2023

PONE-D-21-24749R3 

Co-administration of AYUSH 64 as an adjunct to Standard of Care in mild and moderate COVID-19: A randomized, controlled, multicentric clinical trial 

Dear Dr. Chopra:

I'm pleased to inform you that your manuscript has been deemed suitable for publication in PLOS ONE. Congratulations! Your manuscript is now with our production department. 

Kind regards, 

on behalf of

Dr. Shu-Hsing Cheng 

Guest Editor

PLOS ONE